# Assessment of false discovery rate control in tandem mass spectrometry analysis using entrapment

Bo Wen [1], Jack Freestone[2], Michael Riffle [1], Michael J. MacCoss [1], William S. Noble [1,3] ✉ & Uri Keich [2] ✉

A critical challenge in mass spectrometry proteomics is accurately assessing error control, especially given that software tools employ distinct methods for reporting errors. Many tools are closed-source and poorly documented, leading to inconsistent validation strategies. Here we identify three prevalent methods for validating false discovery rate (FDR) control: one invalid, one providing only a lower bound, and one valid but under-powered. The result is that the proteomics community has limited insight into actual FDR control effectiveness, especially for data-independent acquisition (DIA) analyses. We propose a theoretical framework for entrapment experiments, allowing us to rigorously characterize different approaches. Moreover, we introduce a more powerful evaluation method and apply it alongside existing techniques to assess existing tools. We first validate our analysis in the better-understood data-dependent acquisition setup, and then, we analyze DIA data, where we find that no DIA search tool consistently controls the FDR, with particularly poor performance on single-cell datasets.

In mass spectrometry-based proteomics, controlling the false discovery rate (FDR) among the set of reported proteins, peptides or peptide-spectrum matches (PSMs) is easy to get wrong. Most widely used FDR control procedures in proteomics involve target–decoy competition (TDC) where the observed spectra are searched against a bipartite database comprising real ('target') and shuffled or reversed ('decoy') peptides[1]. Ideally, these procedures would control the actual proportion of false positives among the reported set of discoveries, which is known as the 'false discovery proportion' (FDP); however, in practice, this is impossible because the FDP varies from experiment to experiment and cannot be directly measured[2]. Instead, we control the FDR, which is the expected value of the FDP, that is, its theoretical average over all random aspects of the experiment and its analysis. Although the TDC procedure can be rigorously proven to control the FDR in a spectrum-centric search, subject to several reasonable assumptions[3], in practice, many analysis pipelines implement variants

of the procedure that potentially fail to control the FDR. For example, PSM-level control using TDC is inherently problematic[3,4]. Similarly, most pipelines involve training a semisupervised classification algorithm, such as Percolator[5] or PeptideProphet[6], to rerank PSMs, which in practice can compromise FDR control[7].

Failure to correctly control the FDR can have serious negative implications. Most obviously, if a given analysis pipeline tends to underestimate the FDP—that is, if the pipeline claims that the FDR is controlled, say, at 1%, but the actual average of the FDP is 5%—then the scientific conclusions drawn from those experiments may be invalid. Perhaps more insidiously, invalid FDR control can also impact our choice of analysis pipelines and make comparison of instrument platforms and proteomics workflows impossible. To see why this is the case, consider a hypothetical tool that consistently fails to control the FDR. In a benchmarking experiment, if we compare the number of proteins detected by a collection of analysis tools,

[1]Department of Genome Sciences, University of Washington, Seattle, WA, USA. [2]School of Mathematics and Statistics, University of Sydney, Sydney, New South Wales, Australia. [3]Paul G. Allen School of Computer Science and Engineering, University of Washington, Seattle, WA, USA. ✉e-mail: william-noble@uw.edu; uri.keich@sydney.edu.au

all using a fixed FDR threshold, then the liberally biased tool will have a clear (and unfair) advantage.

To address this concern, therefore, it is important to have a rigorous procedure to evaluate the validity of the FDR control provided by a proteomics analysis pipeline. The standard way to carry out such an evaluation is via an 'entrapment' procedure[8,9], which involves expanding the tool's input dataset so that its search space includes verifiably false entrapment discoveries. Most commonly this is done by expanding the database with peptides taken from proteomes of species that are not expected to be found in the sample, so any such reported peptide is presumably a false discovery. Critically, the distinction between the original input and its entrapment expansion is hidden from the tool itself, so that the entrapment discoveries can subsequently be used to evaluate the tool's FDR control procedure.

Designing an entrapment experiment involves making two major decisions: how the tool's input should be expanded and how the entrapment discoveries should be used to evaluate the tool's FDR control procedure. In this work, we primarily focus on the second decision. This is motivated by a survey of published entrapment experiments, which suggests that, while conceptually simple, correctly carrying out an entrapment estimation can be tricky[10–26]. Indeed, our survey identified a variety of estimation methods, a few of which are invalid as either a lower bound or as an upper bound estimate, while others are often incorrectly used, drawing potentially incorrect conclusions in both cases.

In this work, we expose common errors in existing approaches and introduce a formal framework for entrapment experiments. This framework allows us to rigorously establish properties of existing estimators and to propose a novel entrapment method that allows more accurate evaluation of FDR control for mass spectrometry analysis pipelines. We used entrapment analysis of several popular data-dependent acquisition (DDA) tools to validate that our framework yields results consistent with the field's consensus that these tools generally seem to control the FDR. By contrast, a similar analysis of three popular data-independent acquisition (DIA) tools (DIA-NN[11], Spectronaut and EncyclopeDIA[27]) finds that none of these search tools consistently controls the FDR at the peptide level across all the datasets we investigated. Furthermore, this problem becomes much worse when these DIA tools are evaluated at the protein level. These results suggest an opportunity for the field: insofar as existing methods yield results with unexpectedly high levels of noise, we anticipate that reducing this noise by accurately controlling the FDR has the potential to yield better statistical power in downstream analyses.

## Results

### Many published studies use entrapment incorrectly

Before describing various methods for entrapment analyses, it is important to distinguish between methods that provide estimated upper bounds versus lower bounds of the FDP, and understand their limitations. The primary output of an entrapment procedure can be summarized by plotting the entrapment-estimated FDP as a function of the FDR cutoff used (or reported as a $q$ value) by the evaluated tool. If the entrapment procedure provides an estimated upper bound on the FDP, then the entrapment analysis suggests that the actual FDP falls below the plotted curve. Conversely, the entrapment procedure may provide a lower bound, indicating that the actual FDP falls above the curve. Therefore, applying both an upper bounding and a lower bounding entrapment procedure to a given analysis tool can yield one of three outcomes (Fig. 1): (1) if the upper bound falls below the line $y = x$, then we can take this as empirical evidence suggesting that the tool successfully controls the FDR; (2) conversely, if the lower bound falls above $y = x$, then we can use it as evidence suggesting that the tool fails to control the FDR; (3) if the estimated upper bound is above $y = x$ and the lower bound is below $y = x$, then the experiment is inconclusive.

An important caveat to all of these analyses is that an entrapment procedure aims to gauge the FDP among the discoveries reported by the tool; however, the tool itself is typically designed to control its expected value, the FDR. Hence, we should use reasonably large datasets in our entrapment analysis, where the FDP is typically close to the FDR (by the law of large numbers), or we should average the empirical FDP over multiple entrapment sets, which ameliorates the problem. Even in those cases, the random nature of the FDP implies that some deviations above $y = x$ may be acceptable if they are small and rare.

With that caveat in mind, note that from a statistical perspective, the onus falls on the tool developer to establish that the method indeed controls the FDR, so 'inconclusive' (scenario 3 above) is a tentative strike against the tool. Furthermore, the FDR control should be universal. Consequently, a valid FDR control procedure should achieve scenario 1 for any reasonably large dataset.

We next present the three main approaches in the literature to estimating the FDP. For concreteness, we describe entrapment procedures that expand the peptide database; however, our arguments apply to any type of entrapment expansion, as established rigorously in Supplementary Notes 1 and 2. The first two methods aim to estimate the FDP in the combined, target + entrapment, list of original target and entrapment discoveries, whereas the third method aims to estimate the FDP in a stricter list, focusing only on the original target discoveries and excluding the entrapment discoveries.

The first method, which we refer to as the 'combined' method, estimates the FDP among the target + entrapment ($\mathcal{T} \cup \mathcal{E}_{\mathcal{T}}$) discoveries while taking into account $r$, which is the effective ratio of the entrapment to original target database size

$$\widehat{\text{FDP}}_{\mathcal{T} \cup \mathcal{E}_{\mathcal{T}}} = \frac{N_{\mathcal{E}}(1 + 1/r)}{N_{\mathcal{T}} + N_{\mathcal{E}}}, \tag{1}$$

where $N_{\mathcal{T}}$ and $N_{\mathcal{E}}$ denote the number of original target and entrapment discoveries, respectively. In Supplementary Note 2.2, we prove that, under an assumption analogous to the equal-chance assumption that TDC relies on, equation (1) provides an estimated upper bound, that is, on average it overestimates the true FDP. Thus, the combined method can be used to provide empirical evidence that a given tool successfully controls the FDR among its discoveries (Fig. 1b). Notably, with $r = 1$, equation (1) reduces to Elias and Gygi's original estimation of the FDR in the concatenated target–decoy database, which, in our case, is the target + entrapment database[1]. The combined estimation method has been used to evaluate the DDA analysis tool Mistle[20].

Unfortunately, the combined estimation is often applied incorrectly to establish FDR control after removing the $1/r$ term

$$\underline{\widehat{\text{FDP}}}_{\mathcal{T} \cup \mathcal{E}_{\mathcal{T}}} = \frac{N_{\mathcal{E}}}{N_{\mathcal{T}} + N_{\mathcal{E}}}. \tag{2}$$

The problem is that, as we prove in Supplementary Note 2.1, without the $1/r$ term and assuming that any entrapment discovery is indeed false, equation (2) represents a lower bound on the FDP. As such, this method can only be used to indicate that a tool fails to control the FDR (Fig. 1b), rather than as evidence of FDR control. In what follows we refer to Equation (2) as the 'lower bound.' Table 1 shows that multiple studies incorrectly used the lower bound to validate FDR control, including a recent benchmarking study to evaluate several widely used DIA tools for proteomics and phosphoproteomics DIA data analysis[24]. In that study, the lower bound was used both correctly to point out questionable FDR control, as well as incorrectly as evidence of FDR control. The lower bound has also been used by The et al. to evaluate the 'picked protein' method for FDR control[15], as well as for evaluation of the O-Pair method for glycoproteomics data[12]. However, they both followed the

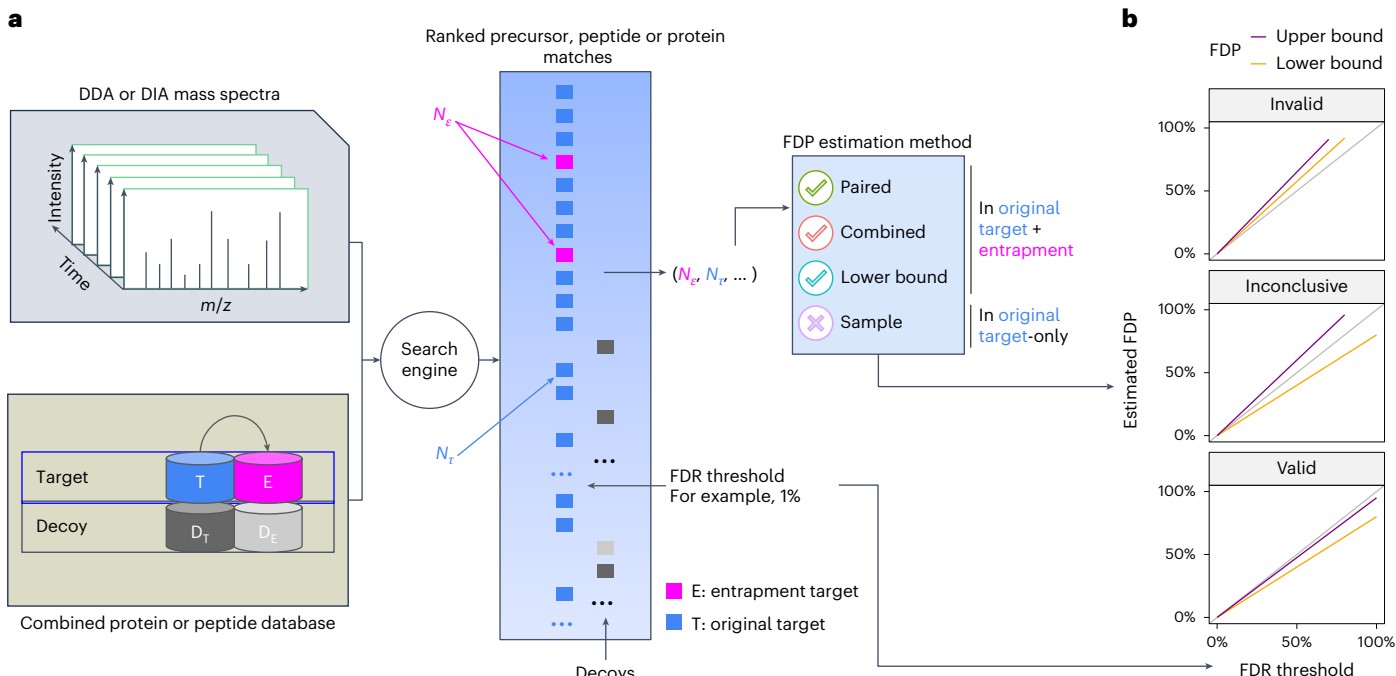

**Fig. 1 | Entrapment strategy for FDR control evaluation. a,** A schematic of a typical entrapment method. The target database is augmented with entrapment sequences, and the augmented database is used by the search tool to produce a ranked list of peptides. In this example, the entrapment is done via the common approach of database expansion and the tool is using decoys to control the FDR, but other methods can be used. The target/entrapment labels are hidden from the search engine but are revealed to the entrapment method, allowing it to provide an estimated FDP. Note that some entrapment estimation methods require additional inputs besides the count of the number of original and entrapment targets. **b,** Comparing the FDR reported or used by a given analysis tool (x axis) to the estimated upper bound (purple) and lower bound (orange) on the FDP produced by two different entrapment estimation methods allows us to conclude that the tool's FDR estimates are valid (bottom) or invalid (top). Middle: if the bounds fall on either side of the line $y = x$, then the analysis is inconclusive.

## Table 1 | Summary of previous entrapment analyses

| Citation | Tools analyzed | DIA | DDA | Entrapment | Entrapment | Valid? |
|---|---|---|---|---|---|---|
| Peckner et al.[10] | Specter | ✓ | | Other | Foreign | |
| Demichev et al.[11] | DIA-NN | ✓ | | Sample | Foreign | |
| Lu et al.[12] | O-Pair | | ✓ | Lower bound* | Foreign | ✓* |
| Sinitcyn et al.[13] | MaxDIA | ✓ | | Sample | Foreign | |
| Lu et al.[14] | DIAmeter | ✓ | | Sample | Shuffled | |
| The et al.[15] | Picked Protein Group FDR | | ✓ | Lower bound* | Shuffled | ✓* |
| Lee et al.[16] | cTDS | | ✓ | Sample | Foreign | |
| Na et al.[17] | Deephos | | ✓ | Sample | Foreign | |
| Lancaster et al.[18] | Spectronaut | ✓ | | Lower bound | Foreign | |
| Scott et al.[19] | GPS | ✓ | | Lower bound | Foreign | |
| Nowatzky et al.[20] | Mistle | | ✓ | Combined | Foreign | ✓ |
| Yu et al.[21] | MSFragger-DIA, DIA-NN | ✓ | | Sample | Foreign | |
| Penny et al.[22] | Spectronaut | ✓ | | Sample | Foreign | |
| Zhang[23] | Mzion | | ✓ | Lower bound | Foreign | |
| Lou et al.[24] | Benchmarking | ✓ | | Lower bound | Foreign | |
| Strauss et al.[25] | AlphaPept | | ✓ | Other | Foreign | |
| Bubis et al.[26] | Spectronaut | ✓ | | Sample | Shuffled | |

The final column indicates whether the entrapment method is deemed invalid to demonstrate FDR control. *Lu et al.[12] and The et al.[15] employed large entrapment databases ($r \geq 5$), so while they used the lower bound, the difference from the combined method was rather small: for example, with $r = 5$ it is 20%.

recommendation of ref. 9 and constructed an entrapment database that was much larger than the original target database. Specifically, they used entrapment databases with $r \geq 5$, where the difference between the combined and lower bound methods would be $1/r \leq 20\%$. Therefore, based on our analysis of the combined method's validity in this work, their evaluation appears to be essentially valid. That said, using a large $r$ means moving further away from the actual intended application of searching just the original target database.

The third approach, which we refer to as the 'sample' estimation method, estimates the FDP only among the original target ($\mathcal{T}$) discoveries as

$$\widehat{\text{FDP}}_{\mathcal{T}} = \frac{N_{\mathcal{E}} \times 1/r}{N_{\mathcal{T}}}, \tag{3}$$

This approach has been employed to evaluate DIA-NN[11], MSFragger-DIA[21], MaxDIA[13] and DIAmeter[14]. We argue that the sample-entrapment method is inherently flawed because, while typically underestimating the FDP, it can also overestimate it in some unusual cases (Supplementary Note 3). Hence, the sample estimation method cannot be used to provide empirical evidence that a tool controls the FDR nor that the tool fails to control the FDR.

Our literature review, summarized in Table 1, indicates that many publications fail to correctly employ the entrapment estimation. Only three of the studies we summarized in the table correctly use entrapment estimation, and all three focused on DDA analysis[12,15,20]. A common mistake is to use the lower bound method, which cannot establish that a given method correctly controls the FDR[18,19,23,24], or to use the problematic sample-entrapment method[11,13,14,21]. Further discussion of some of the studies in Table 1 is provided in Supplementary Note 4.

Note that, in addition to employing different methods to estimate the FDP, the above studies also differ with respect to the entrapment expansion, which, in this case, amounts to how the entrapment database is constructed. In the 'shuffled entrapment' approach, the entrapment sequences are derived analogously to decoy sequences by shuffling the corresponding target sequences, whereas in the 'foreign entrapment' approach they are taken from the proteome of some other species.

## The paired method yields a tighter upper bound on the FDP

As noted above, the combined estimation method provides an estimated upper bound. In practice, we observed that this method can often substantially overestimate the FDP, which motivated us to propose a complementary 'paired estimation' approach. By taking advantage of sample-entrapment pairing information, this method allows us to reduce the conservative bias of the combined method while still retaining its upper bound nature. As such, the paired estimation method is more likely to provide evidence of proper FDR control than the combined method is. For this method to work, say, in peptide-level analysis, we require that each original target peptide be paired with a unique entrapment peptide (so in particular, $r = 1$). In practice, this means that the paired estimation method requires a shuffling or reversal to generate the entrapment peptides.

Given such paired entrapment peptides and still considering peptide-level analysis, the paired method estimates the FDP in the list of target + entrapment discovered peptides by

$$\widehat{\text{FDP}}^{*}_{\mathcal{T} \cup \mathcal{E}_{\mathcal{T}}} = \frac{N_{\mathcal{E}} + N_{\mathcal{E} \geq s > \mathcal{T}} + 2N_{\mathcal{E} > \mathcal{T} \geq s}}{N_{\mathcal{T}} + N_{\mathcal{E}}}, \tag{4}$$

where $s$ is the discovery cutoff score, $N_{\mathcal{E} \geq s > \mathcal{T}}$ denotes the number of discovered entrapment peptides (scoring $\geq s$) for which their paired original target peptides scores $<s$ and $N_{\mathcal{E} > \mathcal{T} \geq s}$ is the number of discovered entrapment peptides for which the paired original target peptides scored lower but were still also discovered. In Supplementary Note 2.3, we recast equation 4 so it can be applicable in our more general entrapment framework. In addition, we introduce the '$k$-matched' generalization of the paired method that, in the case of peptide-level analysis for example, relies on a larger entrapment database, where each target peptide is uniquely associated with $k$ entrapment ones (so $r = k$). Finally, we prove that, under an assumption akin to TDC's equal-chance assumption, both the paired method and its $k$-matched generalization are valid upper bounds in the same averaged sense that the combined method is.

## Comparing estimation methods with controlled experiment data

We first demonstrate the qualitative differences among the above estimation methods—lower bound, sample, combined and paired—using the ISB18 dataset, which consists of DDA data generated from a known mixture of 18 proteins[28]. We used the Tide search engine[29] to carry out FDR control at the peptide level (Methods). Due to the relatively small size of the ISB18 dataset, we averaged each entrapment method's estimated FDP over multiple applications, each with different randomly drawn decoy and entrapment databases. Accordingly, our entrapment methods here are reporting the empirical FDR, that is, the average of the FDP estimates over the 100 drawn decoys and entrapments (Methods).

In the first experiment, the original target database consists of the ISB18 peptides, and the entrapment part consists of shuffled sequences with $r = 1$. We first focus on the paired and combined methods, both of which are estimating the FDP in the same list of target + entrapment discoveries at the given FDR threshold. Notably, the paired method yields an estimated FDR curve that weaves very closely about the line $y = x$, which is firmly within the 95% coverage band of this estimate (Fig. 2a). By contrast, the entire 95% coverage band of the combined estimation lies above the diagonal, demonstrating the upper bound nature of this method. In particular, in this example, we can use the paired method to argue that the FDR seems to be controlled in this case (as we expect it to be), but we cannot make that argument using the combined estimate. As expected, the lower bound curve is below the diagonal but, as such, it is uninformative in this case. Finally, the fact that the sample method is also below the diagonal indicates that, similar to the lower bound, it is probably underestimating the true FDP here.

In the second experiment, we sought to obtain an independent evaluation of the entrapment methods themselves. We did this by taking advantage of the controlled nature in which the ISB18 dataset was generated to conduct a double entrapment experiment, which is designed to gauge how accurately the entrapment methods estimate the FDP. Specifically, we constructed an extended 'original target' database that consisted of the ISB18 peptides augmented by a much larger set of peptides from a foreign species (the castor bean): the ratio of ISB18-to-castor peptides was 1:636. We then applied the paired and sample-entrapment methods using shuffled sequences ($r = 1$) to estimate the FDP among the reported peptides. The controlled nature of the ISB18 dataset implies that any reported castor peptide is a false discovery. At the same time, with a ratio of 1:636 ISB18-to-castor peptides, it is reasonable to assume that any ISB18 reported peptide is a true discovery. Thus, we can directly estimate the FDP in each discovery list and compare it with the estimate produced by the entrapment procedures.

Given the very small proportion of native ISB18 peptides in the extended ISB18 + castor target database, it is not surprising that the combined and paired methods essentially coincide in this setup (Fig. 2b). Notably, though, both provide very accurate estimates of the FDP as confirmed by the independent, 'direct' castor-based estimate, where the latter counts every entrapment or castor peptide as a false discovery. By contrast, the sample entrapment seems to substantially underestimate the castor-based estimate because it is trying to estimate the FDP in the wrong list of discoveries—the ISB18 + castor ones—ignoring the fact that the tool was instructed to control the FDR in the larger set of ISB18 + castor + shuffled entrapments. Finally, the lower bound also underestimates the FDP, but that is not surprising given its definition.

## Entrapment analysis on DDA search engines supports our theory

We next set out to further validate our analysis of the entrapment methods by applying them to gauge the peptide-level FDR control of four well-established DDA search engines—Tide[29], Sage[30], MS-GF+[31] and MSFragger[32]—using data generated from a complex sample, rather

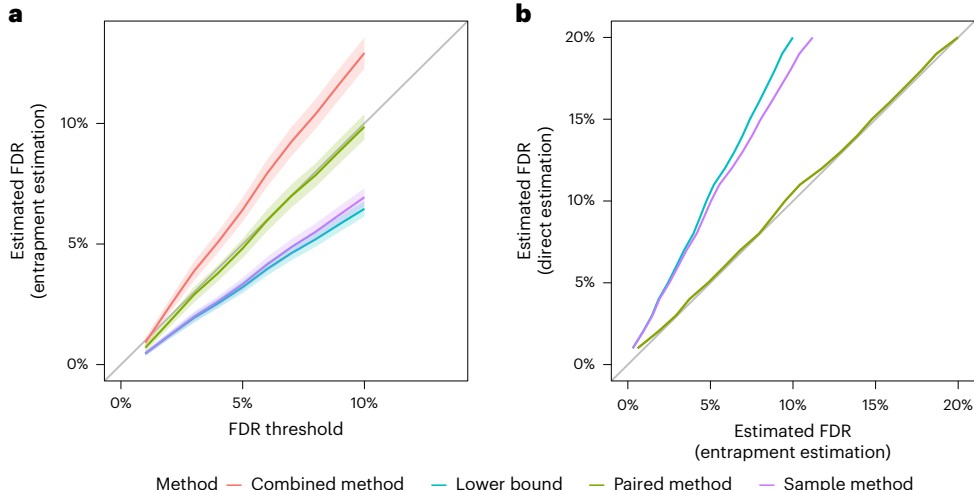

**Fig. 2 | Entrapment analysis using the ISB18 data. a**, The entrapment-estimated FDR or the estimated FDP averaged over 100 sets of decoys and entrapments is plotted (with 95% coverage bands) as a function of the given FDR threshold. The FDP is estimated using four different entrapment methods with the original target database consisting only of the ISB18 peptides. **b**, The estimated FDR or the estimated FDP averaged over 100 sets of decoys and entrapments is plotted against the corresponding castor-based estimates. This experiment uses an original target database consisting of the ISB18 and castor peptides at a ratio of 1:636. The combined and paired curves visually overlap due to the small proportion of native ISB18 peptides in the ISB18 + castor database. The figure summarizes two different types of entrapment experiment, both using shuffled sequences with $r = 1$. The lists of discoveries for **a** and **b** were generated by searching with Tide, followed by peptide-level FDR control.

than from a controlled mixture. For MS-GF+ we carry out peptide-level FDR control using the primary search engine scores, whereas for the other three search engines we use a machine learning post-processor: Percolator-RESET for Tide, Sage's built-in linear discriminant analysis, MSBooster[33] for MSFragger. For this analysis, we use data from 24 tandem mass spectrometry (MS/MS) analyses of the human cell line HEK293, searched against target + entrapment databases for which the human reference proteome was taken as the original target sequences and was paired with shuffled entrapment sequences (so $r = 1$). Thus, in addition to being derived from a more complex sample, this dataset is substantially larger than the ISB18 dataset, as is the original target database. Having established that the sample estimation method is inherently problematic, from here onward we consider only the lower bound, combined and paired estimation procedures.

A priori, given the established nature of DDA FDR analysis, we expect all of these tools to produce valid FDR estimates. Accordingly, the conclusions we draw from the analysis of the HEK293 data (Fig. 3) largely mirror those we drew from the Fig. 2a. Specifically, for all peptide-level analysis tools the paired method yields estimated FDPs that are quite close to the diagonal. The more conservative nature of the combined method is on display again: it is always above the paired estimation curve, with the latter already presenting an upper bound. More specifically, relying on the combined method, we might have reservations on whether we have evidence that Tide+Pecolator-RESET and Sage control the FDR in this setup, but we can argue for such apparent evidence if we use the paired method. As expected, the lower bound seems to consistently substantially underestimate the FDP.

To gauge the possible impact of the inherent variability of the entrapment sequences and, where relevant, the decoy sequences, we again constructed 95% coverage bands for the estimates. Supplementary Fig. 1 shows that, for both Tide and Sage (for which generating these plots was fairly straightforward), the effect of this variability is marginal.

We reach similar conclusions when comparing our entrapment estimation methods using protein-level analysis in DDA data with MaxQuant[34]. Specifically, Supplementary Fig. 2a demonstrates that the paired method provides us with evidence of MaxQuant's successful protein-level FDR control while the combined methods leaves a certain degree of uncertainty about it. At the same time, a comparison of the

panels in Supplementary Fig. 2a,b shows that entrapment analyses based on shuffled and foreign sequences are in fairly good agreement with one another in this case as well.

Finally, we further validated our entrapment analysis by applying the combined, paired and lower bound estimates to peptide-level analysis procedures that do not rely on TDC to control the FDR. Specifically, we analyzed two such procedures in the spirit of the PSM-level one described in ref. 35. Both procedures rely on empirical $P$ values that are estimated from the decoys generated by shuffling the peptides in the target database. The list of discovered peptides is then determined by applying either the Benjamini–Hochberg[2] or Storey's procedure[36] to the list of empirical $P$ values (see Methods for details). To obtain further confidence in our analysis, each entrapment procedure was applied twice, once with shuffled and once with foreign (*Arabidopsis*) entrapment sequences. As expected, in all cases the combined method reported the highest estimated FDP, the lower bound reported the lowest estimate, and the paired method fell between those two (Supplementary Fig. 3). The fact that for both the Benjamini–Hochberg and Storey procedures, the combined estimate is well below the diagonal suggests that both are conservative in this case. This is further suggested by the substantially smaller number of discoveries those two procedures make compared with TDC.

### DIA search engines fail to consistently control the FDR

Turning to tools for analysis of DIA data, we performed a more extensive evaluation. Specifically, we applied three different search engines—DIA-NN, Spectronaut and EncyclopeDIA—to the ten DIA datasets listed in Supplementary Table 1. In the case of EncyclopeDIA, we only analyzed the four datasets for which we have gas phase fractionation runs. In each case, we applied all three entrapment estimation methods separately at the peptide level (for EncyclopeDIA) or precursor level (for DIA-NN and Spectronaut) and at the protein level for each search engine. Precursor-level analysis is analogous to peptide-level analysis except that each discovery is a (possibly modified) peptide and a corresponding charge state. In addition, because Spectronaut only estimates precursor-level FDR for each run, rather than for a full experiment, we only report precursor-level results for a single selected mass spectrometry run for each DIA dataset.

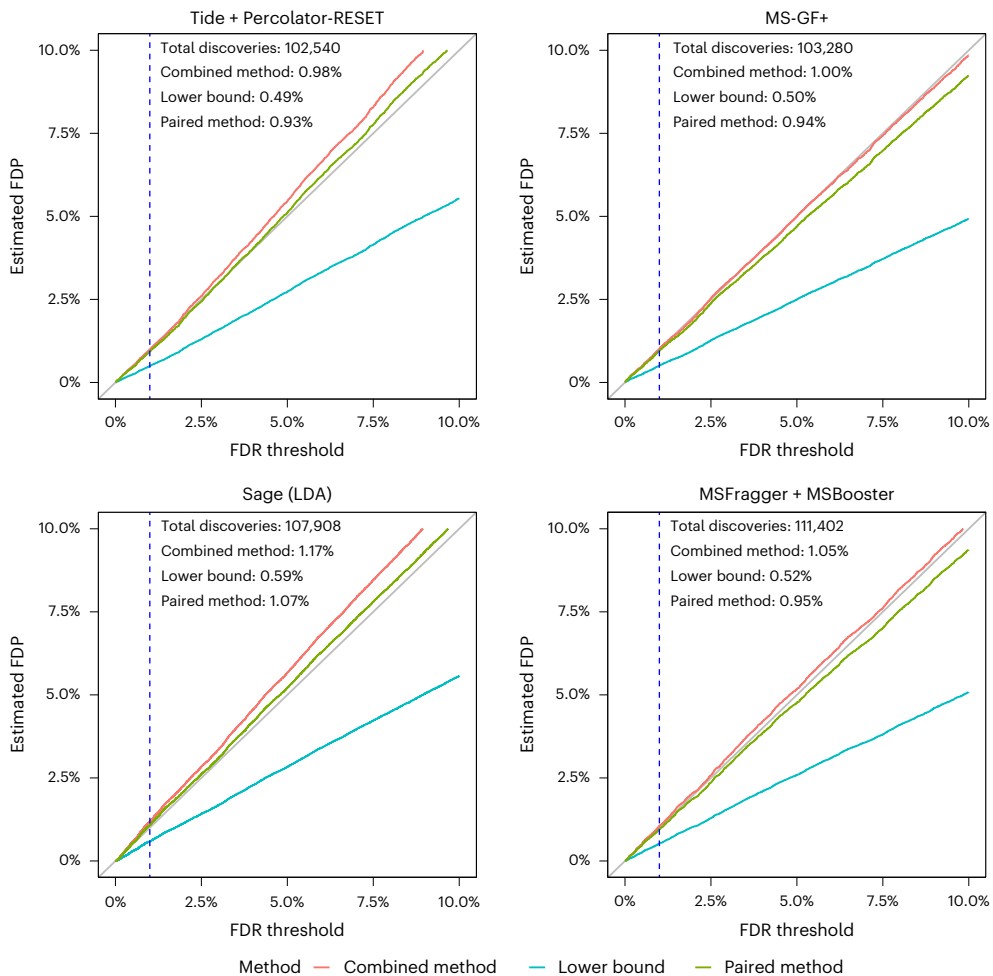

**Fig. 3 | Comparing entrapment procedures using HEK293 DDA data.** Each figure shows, for a given search procedure, the (shuffled with $r = 1$) entrapment-estimated FDP in the combined list of target peptides that was reported at the given FDR threshold. The dashed vertical line is at the 1% FDR threshold, as are the numbers reported in text in the figure. 'LDA' refers to the linear discriminant analysis option in Sage.

The results of this experiment suggest that the precursor or peptide-level FDR control is frequently questionable, whereas protein-level FDR control is frequently invalid (Table 2). For example, for the human-lumos dataset (Fig. 4a) we observe that the precursor or peptide-level FDR control appears to be inconclusive, whereas the protein-level FDR control is apparently invalid: for example, the lower bound on EncylopeDIA's FDP is above 6%. Indeed, keeping in mind that the true FDP is likely to be between the lower bound and the paired-estimated upper bound, we see that the protein-level 1% FDR control appears to be consistently invalid for EncyclopeDIA and Spectronaut and mostly invalid for DIA-NN. Although the peptide-level analysis indicates substantially better control of the FDR, there are still some datasets for which the results are inconclusive or worse. Notably, this is particularly the case for the single-cell (1cell-eclipse) data, where the lower bound on DIA-NN's precursor-level FDP data is above 2.3% and Spectronaut's FDP is above 3.8%. It is worth noting that DIA-NN's and Spectronaut's protein-level estimated FDPs are also highest on that particular dataset. Supplementary Figs. 4–9 complement these observations by providing for all datasets and DIA tools the lower bound, as well as the combined and paired-estimated FDPs for a wide range of FDR thresholds. In particular, the figures consistently demonstrate how the entrapment estimation methods compare with one another, with the combined method reporting the largest estimated FDP, the lower bound the smallest and the paired method in between the other two.

We also performed complementary experiments in which we varied the entrapment-to-original-target ratio $r$. Consistent with

Lou et al. citelou2023benchmarking, we find that the estimated FDP among DIA-NN's reported proteins generally increases with $r$ (Supplementary Figs. 10 and 11). On the other hand, Lou et al. only increased $r$ up to $r \approx 1$ and concluded that DIA-NN controls the FDR; however, because they were using the lower bound, that conclusion is invalid (Supplementary Note 4). By contrast, we explored higher values of $r$ and found that even the lower bound is consistently substantially higher than the corresponding FDR thresholds, indicating quite clearly that DIA-NN apparently fails to control the FDR in those scenarios. For example, for $r = 8$ we see a lower bound of almost 7% (Supplementary Fig. 10). The same phenomenon was observed in the case of Spectronaut as well: for example, the lower bound is above 4% for $r = 6$ (Supplementary Fig. 12). While these observations do not imply that the FDP is as high when searching only the original target database, they do indicate that the tool struggles to control the FDR in some setups. In general, proper FDR control should be applicable to any realistic scenario.

To investigate the practical consequences of erroneous FDR control, we compared the number of discoveries reported at the 1% FDR threshold by DIA-NN to the number of discoveries we get if we use the paired method to guide our cutoff. Specifically, for each of the ten datasets analyzed in Table 2, we asked how many more discoveries DIA-NN reports at the 1% FDR threshold relative to how many discoveries we get when the (paired) entrapment-estimated FDP is at 1%. As shown in Fig. 4b, when the estimated FDP is in the 1–2% range, we see an estimated inflation of up to 6.7% in the number of discoveries at the precursor level and up to 4.7% at the protein level.

**Table 2 | Entrapment analysis of FDR control of three DIA analysis tools**

| Dataset | DIA-NN | | EncyclopeDIA | | Spectronaut | |
|---|---|---|---|---|---|---|
| | Precursor | Protein | Peptide | Protein | Precursor | Protein |
| Human-astral | 0.7–1.3 (?) | 1.5–2.1 (X) | – | – | 0.8–1.5 (?) | 1.6–2.3 (X) |
| Human-qe | 0.7–1.3 (?) | 1.5–2.2 (X) | 0.7–1.3 (?) | 4.7–7.0 (X) | 0.7–1.4 (?) | 1.3–1.9 (X) |
| Human-tripletof | 0.6–1.1 (?) | 1.0–1.5 (X) | 0.7–1.3 (?) | 3.2–4.8 (X) | 0.7-1.4 (?) | 1.5–2.8 (X) |
| Yeast-lumos | 0.6–1.0 (✓) | 1.2–1.4 (X) | 0.4–0.8 (✓) | 4.3–5.2 (X) | 0.8–1.5 (?) | 1.3–1.6 (X) |
| Mouse-qe | 0.7–1.3 (?) | 0.8–1.1 (?) | – | – | 0.7–1.4 (?) | 1.3–1.9 (X) |
| Human-timstof2 | 0.6–1.1 (?) | 0.8–1.2 (?) | – | – | 0.7–1.5 (?) | 1.4–2.1 (X) |
| Human-timstof1 | 0.7–1.2 (?) | 0.8–1.2 (?) | – | – | 0.7–1.3 (?) | 1.1–1.7 (X) |
| 100cell-eclipse | 0.9–1.6 (?) | 1.3–1.8 (X) | – | – | 0.9–1.7 (?) | 1.8–2.9 (X) |
| 1cell-eclipse | 2.3–4.7 (X) | 2.0–3.5 (X) | – | – | 3.8–7.6 (X) | 3.0–5.6 (X) |
| Human-lumos | 0.9–1.7 (?) | 2.1–2.6 (X) | 0.7–1.2 (?) | 6.7–9.1 (X) | 0.7–1.3 (?) | 2.0–3.0 (X) |

Each entry in the table lists the lower bound and paired-estimated upper bound on the empirical FDP among the target+entrapment discoveries reported by the DIA search engine at an FDR threshold of 1%. The entrapment procedures used shuffled entrapment sequences with r=1 and were applied at both the peptide or precursor level and the protein-level. Each entry is followed by an indicator for whether the FDR control for this tool on this dataset is deemed valid (✓), invalid (X) or inconclusive (?). EncyclopeDIA results are provided only for the four datasets with gas phase fractionation data available. Note that the evaluation of each method is based on the full results presented in Fig. 4 and Supplementary Figs. 4–9.

In the case of the single-cell proteomics dataset (1cell-eclipse), the estimated inflation rate is up to 48.3% at the precursor level and 44.2% at the protein level.

## Discussion

Overall, our work identifies a lack of consensus in the field about entrapment estimation methods. Accordingly, we introduce a formal framework within which we can rigorously study entrapment estimation methods. This framework, summarized in Supplementary Table 2, is equally applicable to PSM, peptide and protein-level analyses: provided the assumptions that we specify hold, the corresponding upper or lower bound nature of the estimates is guaranteed (though we generally caution against trying to control the FDR at the PSM level). Moreover, our framework is also applicable to what Madej and Lam referred to as 'entrapment query protocol'[37], where foreign spectra are used to expand the dataset in lieu of the common expansion through a peptide entrapment database.

Applying our entrapment procedures to DIA analysis tools we demonstrate that their FDR control at the peptide or precursor level occasionally seems to fail and typically fails at the protein level. Given the importance of protein-level analysis to mass spectrometry experiments, we believe that this result should serve as a call to further research into this problem. Keep in mind that this is not just a matter of rigorous statistics: often, the detected proteins are then further used to identify differentially expressed proteins, so incorrectly calling the detected proteins can adversely affect our ability to successfully accomplish our ultimate goal.

In general, an upper bound estimate such as the combined estimation can only be used to show valid FDR control, and the lower bound to highlight invalid FDR control. However, if $r$ is large, then the difference between equations (1) and (2) becomes negligible, so each of the methods can be reasonably used for making both arguments. That said, keep in mind that using $r \gg 1$ creates a much larger combined target database, most of which is made of entrapment sequences. Thus, establishing FDR control in this somewhat atypical setup is not as convincing as establishing it for smaller values of $r$ (for example, $r = 1, 2$). Of course, when using smaller values of $r$ we have to take into account that equation (2) is a lower bound and to establish FDR control we need to use the upper bound. Our paired entrapment estimation method, equation (4), and its $k$-matched generalization, Supplementary Equation 3, provide tighter upper bounds than the combined method, particularly for smaller values of $r$ and larger fractions of the native peptides in the original target database.

As mentioned, the validity of an estimate hinges on whether its underlying assumptions are expected to hold, which in turn depends on how the entrapment procedure expands the input dataset. The most common expansion approach in analyzing MS/MS tools is to enlarge the database by adding to it foreign entrapment sequences. In this work, we chose instead to focus on shuffled entrapments because, as we argue in detail in Supplementary Note 5 and Supplementary Figs. 13–16, the use of foreign species raises complex questions associated with the choice of those species; for example, what is the 'right' evolutionary distance for the entrapment species, and whether the entrapment species coincides with a potential source of contamination, as we found in our analysis of the commonly used HEK293 dataset. Such questions will need to be addressed before we can objectively agree on a way to use such foreign entrapment sequences.

Granted, the use of shuffled entrapment is not without its controversy. In particular, Madej and Lam recently argued that using randomly shuffled entrapment sequences to validate tools that rely on shuffled decoys to control the FDR amounts to circular reasoning[37]. While there is merit to this claim, there are many cases where this is not exactly the case.

First, these tools often use the decoy sequences differently than they are used by the entrapment estimation methods. For example, Percolator's cross-validation scheme to improve the ranking of the PSMs can inadvertently misuse the target/decoy label when multiple spectra are generated from the same peptide species[7]. Because in this case the compromised FDR control stems from indirectly peeking at the target/decoy label but not at the target/entrapment label, the problem can be identified even when both the entrapment sequences and the decoys are shuffled. Similarly, there is no circular reasoning when applying shuffled-based entrapments to procedures such as Benjamini–Hochberg and Storey that use shuffled decoys to compute empirical $P$ values.

Second, if we believe, for example, that Assumption 2a in Supplementary Information holds, then the combined estimation method is valid regardless of which type of decoys the analysis tool uses to control the FDR. Note that the field has largely been comfortable with applying TDC that relies on an even stronger assumption than the latter: one that also requires the independence of false discoveries. As previously pointed out, shuffled sequences cannot account for errors due to homologs or more generally due to neighbors (that is, distinct peptides with similar spectra)[38]. However, our analyses here suggest that the same applies when using foreign species such as *Arabidopsis*) in the analysis of a human sample.

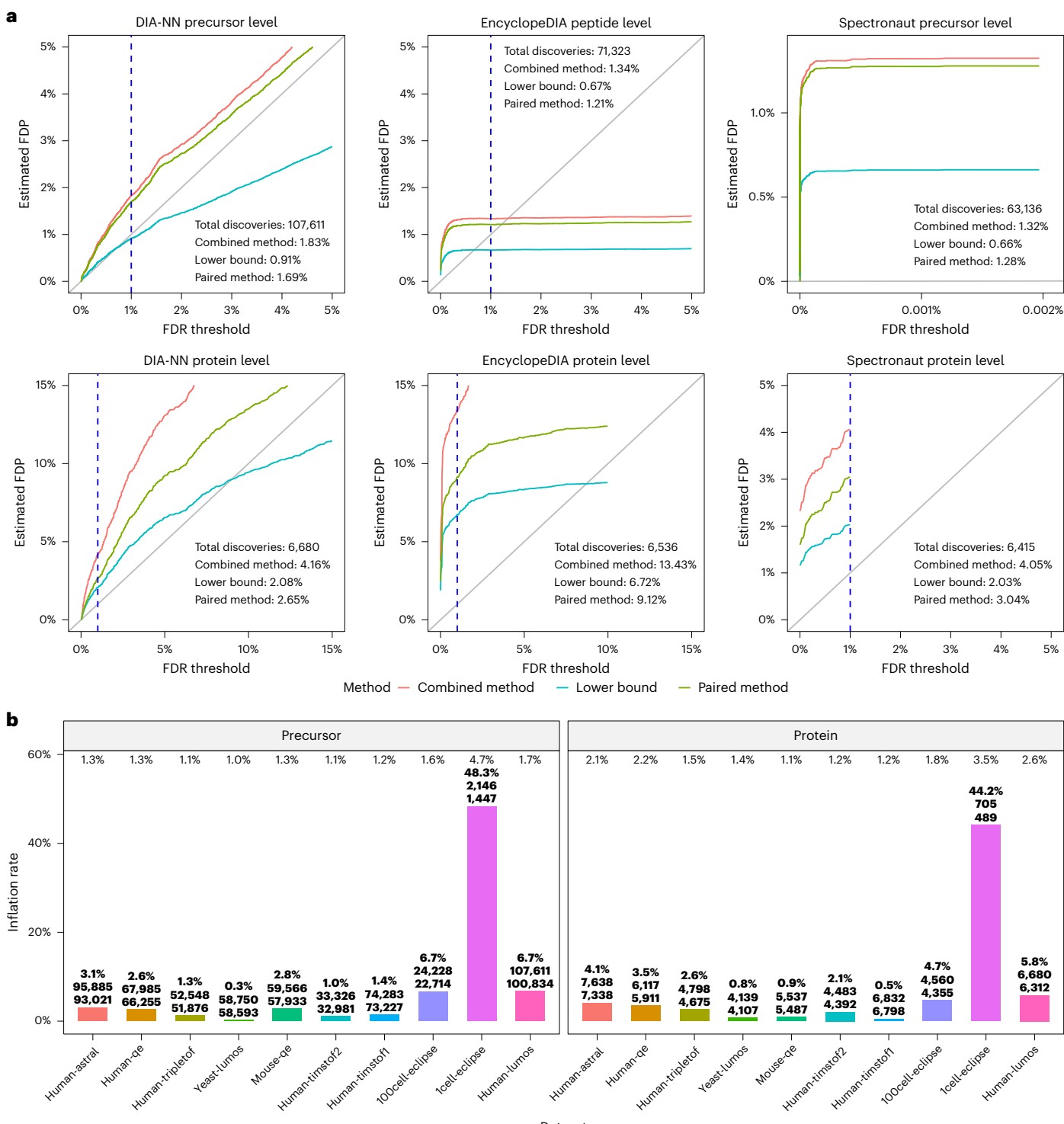

**Fig. 4 | Entrapment evaluation of the FDR control of DIA analysis tools. a**, The FDR control evaluation on the human-lumos DIA dataset. DIA-NN, EncyclopeDIA and Spectronaut were applied to the human-lumos DIA dataset using shuffled entrapment with $r = 1$. The precursor or peptide-level estimated FDPs (top) and the corresponding protein-level analysis (bottom) are shown, with the line $y = x$ included (gray) for reference. In the Spectronaut precursor-level plot, the $x$ axis was set to show the maximum FDR threshold reported by the tool, which was less than the 1% threshold set in the analysis. The dashed vertical lines are at the 1%

FDR threshold, as are the numbers reported in text in **a** and **b**. **b**, Comparing the number of precursor and protein discoveries reported at the 1% FDR threshold by DIA-NN to the inferred number corresponding to the 1% entrapment-estimated FDP (paired method). The numbers at the top are the entrapment-estimated FDPs using the paired method at 1% FDR threshold. The three numbers on each bar are the estimated inflation rate, the reported number of discoveries ($n_1$) at 1% FDR threshold and the entrapment inferred number of discoveries ($n_2$). The estimated inflation rate ($y$ axis) is calculated as $100\% \times (n_1 - n_2)/n_2$.

Indeed, utilizing a mixed foreign and shuffled entrapment approach we find perfect agreement between the shuffled-based combined and paired estimates and the foreign-based one in the controlled

ISB18 setup (Fig. 2b). Similarly, we find little difference when comparing the applications of the combined and lower bound methods to both Tide's and Sage's peptide-level analyses of the HEK293 data

using shuffled sequences and using foreign *Arabidopsis* sequences (Tide: Supplementary Fig. 3, bottom, Sage: Fig. 3, bottom left, and Supplementary Fig. 13, right). The same holds when comparing foreign and shuffled entrapment estimation methods applied to MaxQuant's protein-level analysis (Supplementary Fig. 2), as well as applied to non-TDC-based procedures (Supplementary Fig. 3, first 2 rows).

Moreover, the lower bound method only requires that the entrapment sequences are not present in the sample (Assumptions 1a and 1b in Supplementary Note 2.1). As long as this holds, using this estimate to conclude that a tool apparently fails to control the FDR involves no circular reasoning. In particular, our analysis of DIA tools, which mostly relies on the lower bound estimate, is valid even though we used shuffled sequences.

Regardless, it is clear that the topic of entrapment expansion and its impact on the validity of our entrapment assumptions such as Assumption 1 merits further research. Indeed, considering Madej and Lam's entrapment setup, originally proposed in ref. 39, where foreign spectra are used to expand the dataset in lieu of a peptide entrapment database, one can consider a shuffled-based rather than a foreign-based expansion, for example, by applying the spectrum shuffling protocol of ref. 40. Future research could consider all four possible expansions: shuffled versus foreign and database versus spectrum set, including possibly mixing the strategies as in our double entrapment analysis of the ISB18, as well as using more than one expansion method.

Finally, to facilitate future entrapment analyses, we have produced an open source software tool, FDRBench, that provides two main functions: (1) build entrapment databases using randomly shuffled target sequences or using sequences from foreign species and (2) estimate the FDP using the lower bound, combined and paired methods.

## Online content

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

## Methods

### TDC protocols for controlling the FDR

**TDC.** Initially introduced by Elias and Gygi[1], the following variation of the original TDC procedure was subsequently proved to control the FDR[3]: the input is a list of pairs $(W_i, L_i)$, where $L_i = +1$ for a target PSM, peptide or protein, $L_i = -1$ for a decoy, and $W_i$ is its corresponding score.

The pairs are ordered in decreasing order of their score $W_i$, and for each $k$, we find $D_k$, the number of decoys in the top $k$ pairs and $T = k - D_k$ the number of targets. Given the desired FDR level $\alpha$, TDC sets the rejection/discovery index at

$$K = K(\alpha) := \max\left\{k : \frac{D_k + 1}{\max\{T_k, 1\}} \le \alpha\right\}. \quad (5)$$

Finally, TDC reports all targets among the top $K$ pairs.

Assuming that a pair corresponding to an incorrect discovery is at least as likely to be a decoy as it is to be a target, that is, that $P(L_i = -1) \ge 1/2$ independently of everything else (including of the score $W_i$), TDC rigorously controls the FDR[3,41] among its reported discoveries.

**Peptide-level TDC with PSM-and-peptide.** PSM-and-peptide is a TDC-based procedure for peptide-level analysis introduced in ref. 4. This method involves a double competition: the first is a PSM-level competition where each spectrum is searched against the concatenated target–decoy database, and the best matching peptide is assigned to it (where ties are broken randomly). This defines the (optimal) PSM associated with each spectrum, and then each target or decoy peptide is assigned a score, which is the maximum of the scores of all PSMs that this peptide is part of. PSM-and-peptide introduces a second level of competition to define the scores and labels by utilizing the pairing between each target and its shuffled decoy. Specifically, it keeps only the higher scoring peptide from each target–decoy pair. Any peptide with no matching PSM is assigned the lowest possible score, and all ties are randomly broken. This procedure defines the pair's winning score $W_i$ and its label $L_i = \pm 1$, indicating whether the higher scoring peptide was the target or the decoy. TDC is then applied as above to this list of scores and labels $(W, L)$.

### Controlling the FDR using the Benjamini–Hochberg and Storey

Our peptide-level FDR control using the Benjamini–Hochberg and Storey's procedures began with a Tide target–decoy search without competition. For each target (an original or entrapment target) or decoy peptide, the highest scoring PSM was selected based on the Tailor score[42], and the peptide score was then defined as the score of the selected PSM. Next, a $P$ value was assigned to each target peptide score and was defined as the proportion of decoy peptides with a score as high as the assigned one. For FDR control using Benjamini–Hochberg, the $P$ values of peptides were adjusted using the function p.adjust with the method parameter set as BH (Benjamini–Hochberg) in R. The adjusted $P$ values were then taken as FDR thresholds for downstream analysis. For FDR control using Storey's procedure, the function qvalue from the R package qvalue (version 2.34) was used with its default parameters. The returned $q$ values were then taken as FDR thresholds for downstream analysis.

### Datasets

The MS/MS data used in this study include a wide range of datasets from different vendors, different MS instruments, data acquisition strategies and species. As shown in Supplementary Table 1, a total of 12 datasets were used in the study, including 2 DDA datasets and 10 DIA datasets. All the raw data were downloaded from public databases. The raw data were converted to mgf or mzML format files using MSConvert in ProteoWizard (version 3.0.24031)[43]. For datasets generated using staggered isolation windows, demultiplexing was enabled when using

MSConvert. This excludes the ISB18 data, where ms2 files were obtained from ref. 4. Among the ten DIA datasets, four include gas phase fractionation DIA runs, which were used to build chromatogram libraries for EncyclopeDIA analysis. Two of the datasets were from a previous single-cell proteomics study (dataset ID: PXD023325).

### Entrapment database generation via random sequences

The shuffled entrapment databases were generated differently for precursor-level and peptide-level FDR control evaluation than they were for protein-level analysis. In both cases, the original target protein sequences for human (UP000005640, 20597 proteins), yeast (UP000002311, 6060 proteins) and mouse (UP000000589, 21701 proteins) were downloaded from UniProt (02/2024).

For precursor/peptide-level analysis, the original target proteins were first in silico digested into peptides using trypsin (without proline suppression) with one missed cleavage allowed. The original target peptides database consisted of those with lengths between 7 and 35 amino acids. Then for each original target peptide, we attempted to generate a paired random entrapment peptide as follows. Specifically, the original peptide was shuffled while keeping the C-terminal amino acid fixed and then searched against all original target peptides as well as the previously generated random peptides to ensure it is distinct from all of those. If it was not, we retried to generate such a distinct shuffled peptide up to an additional 20 times. If all those attempts failed we removed the corresponding peptide from the original target database. To generate $r$ matching random entrapment peptides we repeated this shuffling process up to $20 + r$ times to try and obtain $r$ distinct entrapment peptides for each original target peptide. A peptide for which we failed to generate $r$ distinct shuffles after $20 + r$ attempts was removed from the original target database. Supplementary Algorithm 1 summarizes this procedure in pseudocode.

In the protein-level FDR evaluation, for each original target protein, we generated a paired random entrapment protein as follows. We first in silico digested the original protein into peptides using trypsin (without proline suppression). In this step, all the peptides, irrespective of length and mass constraints, were retained and no missed cleavages were considered. Then, for each of these peptides we tried to generate a distinct randomly shuffled entrapment peptide (again, while fixing the C-terminal) as above. This was tried up to 20 additional times for each original peptide, and if all failed to generate a distinct peptide, then (in contrast to the peptide-level analysis) the entrapment peptide associated with the target peptide was identical to the target. Finally, each original peptide in the considered protein was swapped with its randomly generated one. Note that a peptide that appears in multiple proteins or multiple times within a protein was consistently swapped with its uniquely associated paired entrapment peptide in all the proteins it appears. To create an $r$-fold entrapment protein database, we attempted to associate with each digested original target peptide $r$ distinct randomly shuffled entrapment peptides as above. If we failed to do so in $20 + r$ attempted shuffles, then we randomly sampled with replacement $r - n$ additional entrapment peptides from the $n > 0$ distinct shuffles that we managed to generate. If there were no distinct shuffles at all ($n = 0$), then the selected $r$ entrapment peptides were all identical to the original digested peptides. We then used these $r$ entrapment peptides to define $r$ associated entrapment proteins as described above for $r = 1$.

### Entrapment database generation via foreign species sequences

To generate protein-level entrapment databases using proteins from foreign species, a set of proteins from the selected foreign species were randomly selected as entrapment proteins to achieve the desired ratio of $r$ entrapment-to-original target proteins (we used $r \ge 1$).

To generate peptide-level entrapment databases, both original target proteins and the proteins from the foreign species were in

silico digested into peptides using trypsin (without proline suppression) with one missed cleavage allowed. Again, we only considered digested target and entrapment peptides with length between 7 and 35 amino acids. Any foreign species peptides that matched an original target peptide were removed. Finally, we randomly selected as many of the remaining foreign peptides as needed to achieve the desired ratio of $r$ entrapment-to-(remaining)-original target peptides (we used $r \geq 1$). Supplementary Algorithm 2 summarizes this procedure in pseudocode.

Two sets of foreign species were used in this study. The first set of foreign species consisted of *Arabidopsis thaliana* and *Saccharomyces cerevisiae*. The protein sequences from these two species were downloaded from UniProt (02/2024). Specifically, 27448 proteins from *A. thaliana* (UP000006548) and 6060 proteins from *S. cerevisiae* (UP000002311) were used. The second set of foreign species consisted of *Macaca mulatta*, *Callithrix jacchus*, *Papio anubis* and *Mus musculus*. The protein sequences from these species were downloaded from UniProt (primates: 04/2024, mouse: 02/2024). Specifically, 21,590 proteins from *P. anubis* (UP000028761), 21,893 proteins from *M. mulatta* (UP000006718), 22,027 proteins from *C. jacchus* (UP000008225) and 21,701 proteins from *M. musculus* (UP000000589) were used.

### FDR control evaluation procedures

**Using the ISB18 controlled experiment data (with Tide).** We used DDA spectra from an 18-protein mixture, called the ISB18, acquired from a controlled experiment[28]. We implemented our entrapment methods using two different setups: (1) randomly shuffled entrapment sequences and (2) randomly shuffled entrapment sequences in the presence of foreign target sequences from the castor proteome. We used the nine ms2 spectrum files and the castor proteome from ref. 4 and directly downloaded the in-sample protein database 20130710-ISB18-extended.fasta from https://regis-web.systembiology.net/PublicDatasets/database.

Here we used the following variant of the shuffling entrapment protocol described in the 'Entrapment database generation via random sequences' section. For approach 1, we created 100 randomly shuffled databases by digesting the 18-protein database using the tide-index command (default settings) from a recent version of Crux (v4.1.6809338)[44] and randomly shuffling the resulting original target peptides $\mathcal{T}$ while fixing both the C-terminal and the N-terminal amino acids in place. We implemented a narrow search of the combined spectrum files against each of the 100 combined target + shuffled entrapment databases using tide-search (using the automatic fragment and precursor tolerance selection). Next, each entrapment method was used by considering three of the narrow search files, designating one of the randomly shuffled databases as $\mathcal{E}_{\mathcal{T}}$ and the remaining two randomly shuffled databases as the decoys for $\mathcal{T} \cup \mathcal{E}_{\mathcal{T}}$. A small number of randomly shuffled peptides were problematic because they appeared in two different sets: the decoy set of $\mathcal{T}$, $\mathcal{E}_{\mathcal{T}}$ or the decoy set of $\mathcal{E}_{\mathcal{T}}$ and some low complex target peptides were unable to produce three distinct random shuffles. Hence, any target peptide and their corresponding shuffled peptides that contained such a problematic peptide were removed from the search files. Next, we joined the three search files and implemented peptide-level analysis using the PSM-and-peptide protocol with XCorr scores. Finally, we estimated the FDP using each of the entrapment methods. Because we have 100 randomly shuffled databases, we repeated the above analysis 100 times using a different choice of the three narrow search files, ensuring that each narrow search file is considered exactly three times in total. We then estimated the FDR by taking the average of our 100 FDP estimates. The 95% coverage bands that account for the decoy and entrapment sequence variability were computed for each FDR threshold as $\pm 1.96 \times \sigma_n / \sqrt{n}$, where $\sigma_n$ is the standard deviation of the $n = 100$ estimated FDPs at that threshold.

For approach 2, we prepared a new 'original target' peptide database, $\mathcal{T}$, by combining the target sequences digested from the ISB18

protein mixture and the foreign sequences digested from the castor proteome. We then followed the same steps in approach 1 to obtain 100 FDP estimations that were averaged to obtain an FDR estimate using the paired and sample-entrapment methods. We also obtained 100 'direct FDP' estimates, which rely on the number of discovered shuffled entrapment sequences and castor peptides to estimate the number of false discoveries. Specifically, the direct estimate is the ratio of the number of castor and shuffled entrapment discoveries over the total number of (ISB18 + castor + shuffled entrapment) discoveries. These FDP estimates were also averaged to obtain a 'direct FDR' estimate.

**Tide.** To evaluate the FDR control in Tide (within Crux v4.1.6809338)[29] in a more typical setting, the HEK293 DDA MS/MS data were analyzed using Tide. Specifically, a peptide-level paired entrapment database was first generated using the method described in the 'Entrapment database generation via random sequences' section. The Tide-index from the Crux toolkit (https://crux.ms/) was then used to randomly shuffle both the original target peptides and entrapment peptides in the peptide-level entrapment database to obtain decoys (while keeping the C-terminal amino acid fixed) and build an index later used for tide-search. Carbamidomethylation of cysteine was set as a fixed modification, and no variable modification was used in this step. Next, the HEK293 MS/MS data were searched against the combined database generated in the previous step using tide-search from the Crux toolkit with the following parameters: tailor-calibration, enabled; Sp scoring, enabled; precursor ion mass tolerance, 20 ppm; fragment tolerance, 0.02 $m/z$; fragment offset, 0. All other Tide parameters were set as default. The FDR control of the Tide search result at the peptide level was performed using the single-decoy Percolator-RESET (version 0.0.6)[45] with specifying the Tailor score[42] as the primary score (all other parameters were set as default).

**Sage.** To evaluate the peptide-level FDR control in Sage (version 0.14.6)[30], we generated the same target + shuffled entrapment database as described above for Tide. In addition, to demonstrate the evolutionary distance problem with foreign entrapment, we also used Sage with foreign peptide entrapments as described in section 'entrapment database generation via foreign species sequences'. In both cases, the HEK293 dataset was searched against the target + entrapment database using Sage with the following parameters: fixed modification, carbamidomethyl (C); no variable modifications; precursor ion mass tolerance, 20 ppm; fragment ion tolerance, 20 ppm; enzyme digestion was disabled; peptide length range, 7–35; isotope error range was set to '[0,0]'. All other parameters were set as default. Sage's built-in FDR control procedure was used. The 'peptide_q' from Sage's output was used as the peptide $q$ value for downstream analysis.

**MS-GF+.** To evaluate the FDR control in MS-GF+ (version 2023.01.12)[31], a peptide-level entrapment database was used that contained sample peptides, paired entrapment peptides and their paired decoy peptides. The peptide-level entrapment database was generated using the method described in the 'Entrapment database generation via random sequences' section. Specifically, in generating the peptide database, three different random peptides were generated for each target peptide, one decoy peptide was taken as paired entrapment peptide for the target while the other two decoy peptides were taken as decoy peptides. The HEK293 dataset was searched against the entrapment database using the following parameters: fixed modification, carbamidomethyl (C); no variable modifications; precursor ion mass tolerance, 20 ppm; range of allowed isotope peak errors, '0,0'; peptide length range, 7–35; instrument ID, 3 (Q-Exactive); fragmentation method, 3 (HCD); protocol ID, 5 (standard); N-terminal methionine cleavage was disabled. No enzyme digestion was applied. The parameter '-tda' was set to 0 to allow using the decoy peptides contained in the peptide database.

All other parameters were set as default. The 'PepQValue' from the output of MS-GF+ was used as the peptide *q* value for downstream analysis.

**FragPipe.** To evaluate peptide-level FDR control of the FragPipe pipeline (version 21.1)[21,32,33] on DDA data, the HEK293 dataset was searched against the same peptide entrapment database used in the MS-GF+ peptide-level FDR control analysis. The 'Default' workflow setting in FragPipe was used with several parameters changed as follows. The enzyme digestion was configured to disable in silico digestion. The setting of 'Clip N-term M' was disabled. No variable modification was used. The isotope error was set to zero. The calibration and optimization setting was disabled. The 'pin' files generated by MSBooster[33] were combined and used for Percolator (version 3.6.4)[5] analysis. The peptide-level result from Percolator was then used for downstream analysis.

**MaxQuant.** To evaluate protein-level FDR control of MaxQuant (version 2.6.5.0)[34] on DDA data, the raw files of the HEK293 dataset were searched against two protein-level entrapment databases separately using the following parameters: fixed modification, carbamidomethyl (C); variable modifications, oxidation (M); the default enzyme 'Trypsin/P' was used with a maximum of one missed cleavage site allowed; the setting of 'Include contaminants' was disabled; protein-level FDR threshold was set as 0.1. All other parameters were set to their default values. The first entrapment database was generated using the random shuffling method described in the 'Entrapment database generation via random sequences' section with *r* = 1. The second entrapment database was generated using the method described in section 'entrapment database generation via foreign species sequences', in which the proteins from *A. thaliana* were used as entrapment proteins with *r* = 1. The protein files 'proteinGroups.txt' generated by MaxQuant were used for downstream analysis. The 'Q-value' from the files was used as protein *q* value for downstream analysis.

**DIA-NN.** For precursor-level FDR control evaluation of DIA-NN (version 1.8.1)[11], a peptide-level entrapment database, was used that contained the original target and their paired entrapment peptides for each DIA dataset. Each peptide-level entrapment database was generated using the method described in the 'Entrapment database generation via random sequences' section. DIA-NN analysis was performed using the following parameters: fixed modification, carbamidomethyl (C); no variable modifications; enzyme digestion was disabled; peptide length range, 7–35; precursor charge range, 2–4. The setting of 'N-term M excision' was disabled. The precursor FDR was set to 10%. All other parameters were set to their default values. For single run DIA data, the 'Q.Value' from the main report was used as precursor *q* value for downstream analysis. For datasets with multiple runs, the 'Lib.Q.Value' from the main report was used as precursor *q* value for downstream analysis.

For protein-level FDR control evaluation, we ran DIA-NN in its library-free mode using an entrapment database as described in the 'Entrapment database generation via random sequences' section. The enzyme and peptide length settings were the same as peptide-level entrapment database generation in the precursor-level FDR control evaluation. The precursor FDR threshold was set to 1%. All other parameters were set as the same with the precursor-level analysis. For single run DIA data, the 'PG.Q.Value' from the main report was used as the protein *q* value for downstream analysis. For datasets with multiple runs, the 'Lib.PG.Q.Value' from the main report was used as the protein *q* value for downstream analysis. If a protein group included ≥2 proteins and at least one of them was from the original target database, it was taken as an original target protein group.

**EncyclopeDIA.** We evaluated both the peptide-level and the protein-level FDR control of the gas phase fractionated (GPF) chromatogram library analysis workflow with EncyclopeDIA (version 2.12.30)[27,46].

We first used Oktoberfest (version 0.6.2) with Prosit models (fragment ion intensity prediction model: Prosit_2020_intensity_HCD, retention time model: Prosit_2019_irt)[47,48] to generate two in silico spectral libraries for each DIA dataset that were later used for EncyclopeDIA analysis. In the spectral library generation step, carbamidomethyl of cysteine was considered as a fixed modification, and no variable modifications were considered. Precursor charges 2–4 were considered. The normalized collision energy parameter was set to 27. The first spectral library was used for peptide-level FDR control analysis in which the input to Oktoberfest for library generation was a peptide database in csv format containing the target peptides and their paired entrapment peptides as described in the 'Entrapment database generation via random sequences' section. The second spectral library was used for protein-level FDR control analysis in which the input to Oktoberfest for library generation was a protein database containing the target proteins and their paired entrapment proteins as described in the 'Entrapment database generation via random sequences' section. For the second library generation for each DIA dataset, trypsin (without proline suppression) with one missed cleavage allowed was used and only peptides with lengths between 7 and 35 amino acids were considered in Oktoberfest. The methionine cleavage was disabled by making a minor change to the function of digest in Oktoberfest.

Next, for each FDR control evaluation analysis (peptide level or protein level), we generated a new spectral library by searching a set of GPF library DIA runs against the corresponding in silico spectral library using EncyclopeDIA.

Finally, we searched quant DIA runs against the GPF-derived spectral library in each FDR control evaluation analysis. In the analysis, the V2 scoring of EncyclopeDIA was enabled except for the TripleTOF 5600 dataset (PXD028735, human-tripletof). For protein-level FDR evaluation, the protein FDR threshold in the quant DIA analysis was set to 10% and the peptide FDR threshold was set to 1%. If a protein group included ≥2 proteins and at least one of them was from the original target database, it was taken as an original target protein group. For peptide-level FDR evaluation, the protein FDR threshold in the quant DIA analysis was set to 1%, and the peptide FDR threshold was set to 10%. The EncyclopeDIA analysis was run through the nf-skyline-dia-ms workflow (https://nf-skyline-dia-ms.readthedocs.io).

**Spectronaut.** We evaluated both the precursor-level and the protein-level FDR control of the library-free analysis workflow (directDIA) in Spectronaut (version 18.7.240325.55695).

For evaluating the protein-level FDR control we constructed a protein database containing the original target proteins and their paired entrapment proteins as described in the 'Entrapment database generation via random sequences' section. We used Spectronaut with the following settings: enzyme specificity, trypsin (without proline suppression); maximum missed cleavages, 1; peptide length range, 7–35; toggle N-terminal M, disabled. All other parameters were set as default. We used the 'PG.Qvalue' from Spectronaut output as the protein *q* value for downstream analysis. If a protein group included ≥2 proteins and at least one of them was from the original target database, it was taken as an original target protein group.

For evaluating the precursor-level FDR control we constructed a peptide database containing the target peptides and their paired entrapment peptides as described in the 'Entrapment database generation via random sequences' section. Because Spectronaut only estimates precursor-level FDR for each run, this analysis was done using only one MS run from each DIA dataset. No variable modification was set and no enzyme digestion was applied. The peptide length range was set from 7 to 35; toggle N-terminal M was disabled. In addition, the precursor PEP cutoff, protein *q* value cutoff (experiment and run), protein PEP cutoff were set to 0.99. All the other Spectronaut parameters were set as default. We used the 'EG.Qvalue' from Spectronaut output as the precursor *q* value for downstream analysis.

**Reporting summary**

Further information on research design is available in the Nature Portfolio Reporting Summary linked to this article.

## Data availability

The MS/MS datasets used in this study were all downloaded from public databases. The ISB18 DDA data were download from https:// regis-web.systemsbiology.net/PublicDatasets/18_Mix/Mix_7/ ORBITRAP/RAW_Data/. The HEK293 DDA data were download from PRIDE with the accession number PXD001468. The DIA datasets were downloaded from Panorama, PRIDE, jPOST or MassIVE through the following accession numbers: PXD042704, PXD034525, PXD028735, PXD041421, PXD017703, PXD023325, PXD012988 and MSV000084000. The source data for reproducing the figures are available via Zenodo at https://doi.org/10.5281/zenodo.15073580 (ref. 49).

## Code availability

We implemented the entrapment database generation methods as well as the different FDP estimation methods in a Java tool called FDRBench. The source code is available with an Apache license via GitHub at https://github.com/Noble-Lab/FDRBench. The scripts for reproducing the figures are available via GitHub at https://github.com/ Noble-Lab/FDRBench_manuscript.

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

## Acknowledgements

We gratefully acknowledge F. Yu for helpful discussions. This work was supported by the National Science Foundation (award no. 2245300, W.S.N.), National Institutes of Health award R24 (no. GM141156, M.J.M.), the Intelligence Advanced Research Projects Activity (IARPA) TEI-REX program (contract no. W911NF2220059, M.J.M.), the National Science Foundation Graduate Research Fellowship Program (grant no. DGE-2140004, B.W.) and the University of Washington's Proteomics Resource (grant no. UWPR95794, M.R.). The views and conclusions contained should not be interpreted as necessarily representing the official policies, either expressed or implied, of ODNI, IARPA, ARO or the US Government. The US Government is authorized to reproduce and distribute preprints for governmental purposes notwithstanding any copyright annotation therein. The funders had no role in study design, data collection and analysis, decision to publish or preparation of the manuscript.

## Author contributions

B.W., U.K., W.S.N. and M.J.M conceived the study. B.W. developed the software. B.W. and J.F. performed the data analysis. M.R. made updates to the DIA workflow nf-skyline-dia-ms to facilitate the analysis. B.W., U.K., W.S.N. and J.F. wrote the paper with input from M.J.M. All authors read and approved the final paper.

## Funding

## Competing interests

The MacCoss Lab at the University of Washington receives funding from Agilent, Bruker, Sciex, Shimadzu, Thermo Fisher Scientific, and Waters to support the development of Skyline, a quantitative analysis software tool. M.J.M. is a paid consultant for Thermo Fisher Scientific. The other authors declare no competing interests.

## Additional information

**Correspondence and requests for materials** should be addressed to William S. Noble or Uri Keich.

# Reporting Summary

## Statistics

For all statistical analyses, confirm that the following items are present in the figure legend, table legend, main text, or Methods section.

| n/a | Confirmed | |
|---|---|---|
| ☒ | ☐ | The exact sample size (*n*) for each experimental group/condition, given as a discrete number and unit of measurement |
| ☒ | ☐ | A statement on whether measurements were taken from distinct samples or whether the same sample was measured repeatedly |
| ☒ | ☐ | The statistical test(s) used AND whether they are one- or two-sided *Only common tests should be described solely by name; describe more complex techniques in the Methods section.* |
| ☒ | ☐ | A description of all covariates tested |
| ☐ | ☒ | A description of any assumptions or corrections, such as tests of normality and adjustment for multiple comparisons |
| ☒ | ☐ | A full description of the statistical parameters including central tendency (e.g. means) or other basic estimates (e.g. regression coefficient) AND variation (e.g. standard deviation) or associated estimates of uncertainty (e.g. confidence intervals) |
| ☒ | ☐ | For null hypothesis testing, the test statistic (e.g. *F*, *t*, *r*) with confidence intervals, effect sizes, degrees of freedom and *P* value noted *Give P values as exact values whenever suitable.* |
| ☒ | ☐ | For Bayesian analysis, information on the choice of priors and Markov chain Monte Carlo settings |
| ☒ | ☐ | For hierarchical and complex designs, identification of the appropriate level for tests and full reporting of outcomes |
| ☒ | ☐ | Estimates of effect sizes (e.g. Cohen's *d*, Pearson's *r*), indicating how they were calculated |

*Our web collection on statistics for biologists contains articles on many of the points above.*

## Software and code

Policy information about availability of computer code

| Data collection | No custom software was used to collect the data in this study. |
|---|---|
| Data analysis | Data in this study was processed and analyzed using the following software tools:<br>- Tide (within Crux v4.1.6809338)<br>- Sage (version 0.14.6)<br>- MS-GF+ (version 2023.01.12)<br>- FragPipe (version 21.1)<br>- MaxQuant (version 2.6.5.0)<br>- DIA-NN (version 1.8.1)<br>- EncyclopeDIA (version 2.12.30)<br>- Spectronaut (version 18.7.240325.55695)<br>- Oktoberfest (version 0.6.2)<br>- R (version 4.3.1)<br>Additionally, FDRBench's source code is available under the Apache 2.0 license at https://github.com/Noble-Lab/FDRBench. |

For manuscripts utilizing custom algorithms or software that are central to the research but not yet described in published literature, software must be made available to editors and reviewers. We strongly encourage code deposition in a community repository (e.g. GitHub). See the Nature Portfolio guidelines for submitting code & software for further information.

## Data

Policy information about availability of data

All manuscripts must include a data availability statement. This statement should provide the following information, where applicable:

- Accession codes, unique identifiers, or web links for publicly available datasets
- A description of any restrictions on data availability
- For clinical datasets or third party data, please ensure that the statement adheres to our policy

The MS/MS datasets used in this study were all downloaded from public databases. The ISB18 DDA data were download from https://regis-web.systemsbiology.net/PublicDatasets/18_Mix/Mix_7/ORBITRAP/RAW_Data/. The HEK293 DDA data were download from PRIDE with the accession number PXD001468. The DIA datasets were downloaded from Panorama, PRIDE, jPOST or MassIVE through the following accession numbers: PXD042704, PXD034525, PXD028735, PXD041421, PXD017703,PXD023325, PXD012988 and MSV000084000. The source data for reproducing the figures is available at https://doi.org/10.5281/zenodo.15073580.

## Human research participants

Policy information about studies involving human research participants and Sex and Gender in Research.

| Reporting on sex and gender | Not applicable |
| Population characteristics | Not applicable |
| Recruitment | Not applicable |
| Ethics oversight | Not applicable |

Note that full information on the approval of the study protocol must also be provided in the manuscript.

# Field-specific reporting

Please select the one below that is the best fit for your research. If you are not sure, read the appropriate sections before making your selection.

☒ Life sciences ☐ Behavioural & social sciences ☐ Ecological, evolutionary & environmental sciences

For a reference copy of the document with all sections, see nature.com/documents/nr-reporting-summary-flat.pdf

# Life sciences study design

All studies must disclose on these points even when the disclosure is negative.

| Sample size | No new data was generated for this study. All the datasets used in the study were downloaded from public databases, so the sample sizes were determined by the authors of the original studies. In this study, we selected MS runs for use to cover a single MS run scenario to multiple MS runs scenario. |
| Data exclusions | For the ISB18 DDA dataset, we used 9 of the 10 raw files in the present study. We excluded one raw file due to file size issue. |
| Replication | All the datasets used in the study were downloaded from public databases. Some of the datasets used in the study include replicates. |
| Randomization | All the datasets used in the study were downloaded from public databases. We selected datasets to cover different MS instruments and data types. |
| Blinding | Blinding is not relevant because the study focuses on method development. The study doesn't involve any new data generation. |

# Reporting for specific materials, systems and methods

We require information from authors about some types of materials, experimental systems and methods used in many studies. Here, indicate whether each material, system or method listed is relevant to your study. If you are not sure if a list item applies to your research, read the appropriate section before selecting a response.

## Materials & experimental systems

| n/a | Involved in the study |
|-----|----------------------|
| ☒ | ☐ Antibodies |
| ☒ | ☐ Eukaryotic cell lines |
| ☒ | ☐ Palaeontology and archaeology |
| ☒ | ☐ Animals and other organisms |
| ☒ | ☐ Clinical data |
| ☒ | ☐ Dual use research of concern |

## Methods

| n/a | Involved in the study |
|-----|----------------------|
| ☒ | ☐ ChIP-seq |
| ☒ | ☐ Flow cytometry |
| ☒ | ☐ MRI-based neuroimaging |

