## [Peer Review File · Nature Methods]

Assessment of false discovery rate control in tandem mass spectrometry analysis using entrapment

Corresponding Author: Dr Uri Keich

A version of this paper was originally rejected for publication by Nature Methods, however that decision was reconsidered after appeal by the authors.

Version 0:

Decision Letter:

21st Aug 2024

Dear Dr. Keich,

Your Article entitled "Assessment of false discovery rate control in tandem mass spectrometry analysis using entrapment" has now been seen by 3 reviewers, whose comments are attached. While they find your work of potential interest, they have raised serious concerns which in our view are sufficiently important that they preclude publication of the work in Nature Methods, at least in its present form.

As you will see, the reviewers raise several technical concerns about the approach, the presentation of method and its applicability and utility.

Should further experimental data allow you to fully address these criticisms to ensure the technical points are sufficiently alleviated, we would be willing to look at a revised manuscript (unless, of course, something similar has by then been accepted at Nature Methods or appeared elsewhere). This includes submission or publication of a portion of this work somewhere else. We hope you understand that until we have read the revised paper in its entirety we cannot promise that it will be sent back for peer-review.

If you are interested in revising this manuscript for submission to Nature Methods in the future, please contact me to discuss your appeal before making any revisions. Otherwise, we hope that you find the reviewers' comments helpful when preparing your paper for submission elsewhere.

Sincerely,
Arunima

Arunima Singh, Ph.D.
Senior Editor
Nature Methods

Reviewers' Comments:

Reviewer #1:

Remarks to the Author:

In this paper, Wen et al. focus on the correct way to validate FDR control tools applied to the peptide identification problem in proteomics, using benchmarking methods build on the concept of "entrapment database". The contributions are multiple and important:

1- They provide actionable guidelines intended to the computational proteomic experts, to avoid the entrapment concept is used incorrectly, or implemented into a flawed procedure.

General comment: In my opinion, this contribution is important to very important, considering the current lack of rigor in the field, but it addresses a niche readership (essentially proteomic software developers). More critically, this contribution is not presented in a sufficiently formal way, so that it may miss its goal. Note that here, "insufficiently formal" does not refer to the statistical correctness of the ideas exposed (which from my viewpoint is excellent), but to how it should be concretely applied by non-experts (points of vigilance, limitations, precise itemization of the recipe, risk of "circular reasoning" as reported by Madej and Lam [25], etc.). Therefore, I believe improving the manuscript with this respect should essentially require adjusting the presentation (see details below).

2- They propose a new (tighter) upper bound on the FDR (plus the associated software).

General comment: The proposal is theoretically sounded, while carefully evaluated on the empirical side, notably using a clever double-entrapment strategy. Likewise, providing software to improve future evaluation is valuable. However, there is one blind spot of importance, which relates to the same issue as the point above, namely the possible limits of the evaluation owing to the risk of circular reasoning (broadly speaking, "I assume the equal chance assumption (ECA) holds between mismatches and matches onto shuffled sequences. Therefore TDC-based FDR should be correct, but to empirically demonstrate it, I will rely on an entrapment approach which requires the exact same assumption", see [25]). I do not doubt on the correctness of the new bound (the theoretical support is solid) and I am positive about this contribution, but I think an orthogonal empirical validation is nevertheless necessary.

3- They expose the lack of rigor about how FDR control was validated in the currently used tools, specially DIA ones.

General comment: The (dramatic) conclusion is very timely, extremely interesting, and calls for a mass reaction in the proteomics community, considering the recent hype about using DIA approaches, notably for single-cell proteomics. Based on the manuscript, it is tempting to assume that the main reason why DIA tools are preferred for SC proteomics is because a less strict FDR control makes it possible to have longer lists of identified peptides, regardless the DDA vs. DIA data quality. As is, this contribution should have a long-standing echo in the community. I am totally supporting the authors' analysis, and I wonder about making it more explicit in the title or the abstract of the paper (BTW, I found the abstract too long and too technical and I assume a catchier line would better support the paper... Unfortunately, I have no concrete proposal to help the authors in this direction yet).

Overall, I support this article and I would be pleased to see it published in Nature Methods in the end. However, I am partially unsatisfied with some aspects of the current version, in relation to points 1 and 2 above. In summary, this paper has the potential to close a decade-old wandering in the computational proteomics community with respect to the correct use of entrapment sequences, and this must be acknowledged. However, this wandering originated in the fuzziness of the earlier papers about entrapment, as purposely highlighted by Madej and Lam [25] (BTW, there is a typo in the paper, Majed instead of Madej): the original lack of precise definitions and of hypotheses led the possibility to elaborate sophisticated entrapment-based procedure, more or less correct, and implicitly rooted on circular reasoning. To avoid perpetuating this, I think it is important the authors provide the clearest possible guidelines and accept discussing in more details the limitations of the approach. Concretely:

1- Provide a more formal and more precise definition of entrapment:

- In [25], Madej and Lam distinguish different uses or types of entrapment. Does the authors agree on this classification? And which type of entrapment they focus on? (e.g., as pointed in supp mat, the authors prefer shuffling over distant organism, but this should be stated formally and earlier, as to better delineate the scope).
- Likewise the authors of [25] root the entrapment concept into publications earlier than [10]. This may not old depending on the type of entrapment approaches the authors focus one, but still, clarification is needed from the readership viewpoint, as to delineate which methods the proposed recommendations apply to.

2- Provide a more formal definition of the hypotheses entrapment is based on and discuss whether decoys and entrapment sequences are exactly the same or not (hypotheses for correct generation, and how they should be used) as this distinction is at the core of a possible circular reasoning.

3- Provide a more formal description of the entrapment implementation (like with a pseudo-algorithm for instance?).

4- I believe the authors overlook the circular reasoning highlighted by Madej et al. Notably they say: "We only partially agree with this statement. Specifically, we contend that this reservation is valid only if the analysis tool's use of the decoy sequences is restricted to estimating the FDR using the same estimation method that the entrapment relies on." I agree the case highlighted is problematic. However, contrarily to what the authors claim, it is not the only possible case: Depending on how the shuffle sequences (be them entrapment ones or decoys) distribute (because of a flawed generation process or an artifact that affects both procedures) and how this impacts the competition against real sequences, the entrapment validation may be equally flawed, despite relying on a different mechanism. Here are 3 examples:

* Percolator: In addition to the issue pointed in [8], percolator can learn to distinguish shuffled sequences from target ones [<https://doi.org/10.1021/acs.jproteome.8b00991>], leading to prefer the latter ones to the former ones (in other words, the ECA no longer holds), and thus to underestimating the FDR. If shuffled sequences are also used for the entrapment validation, the same problem will lead to hiding the underestimated FDR, as percolator will recognize the entrapment sequences as easily as the decoy ones.

* Proteogenomics: In [<https://doi.org/10.1186/s13059-022-02701-2>], we have empirically demonstrated that the reduction of the target database size according to transcriptomic assays can break the ECA (briefly, because in the process, the best decoy score is lowered much more than the best target one, leading to an imbalanced competition). How entrapment would be affected by this (as it also requires the ECA to hold) is not clear. I do not think it is possible to make concrete conclusions without investigating this, and as the author say, the onus fall on the developer.

* We have shown the same with the reduction of the precursor mass tolerance windows

[<https://doi.org/10.1021/acs.analchem.0c00328>]. We observed this phenomenon did not affect target-decoy without competition, and I personally believe (even though I have not been able to demonstrate it so far) that subsetting the databases (using the precursor windows or the transcriptomic data) leads to breaking the ECA. To support this intuition, consider the case of only 1 target sequence T falling in the windows: it will be selected as the best target regardless its score. If T is incorrect but almost correct (eg, only a couple of AA permutations) the probability that its shuffled version outscores T is much smaller than if several shuffled sequence would have fallen in the mass tolerance windows. These phenomena are not considered in [11] (notably Theorem 2), as the ECA is taken for granted whatever the TDC setting. However, considering the empirical evidence, it is legitimate to acknowledge the ECA may sometime not hold.

To conclude, any situation where the ECA may not hold any longer, be it because of how the data are processed, of how the FDR is controlled in the TDC procedure, or of how sequences are shuffled, can lead to flawed entrapment procedures. Finally, there are many situations beyond what the authors contend where the circular reasoning is a risk, and this should be better discussed and delineated, notably thanks to more formal presentations of the definitions, hypotheses, implementations, etc. (as commented above). This said, I acknowledge that in the many cases where the ECA holds, entrapment as presented by the authors is valid, so my point is not to minimize the nice results of the paper, but only to better delineate their validity domain.

5- The previous point explains why I found a blind spot in the empirical evaluation of the new tighter upper bound on the FDP. Considering competition against shuffle sequences are used both for the validation (entrapment) and the FDR control (target-decoy competition), circular reasoning cannot be excluded. The only way to overcome this is to propose an orthogonal validation, like for instance:

- * Target-decoy without competition

- * Benjamini-Hochberg procedure

- * Empirical null as proposed in early papers by Nesvizhskii.

- * Conditional randomization test: essentially the same idea the authors developed in

[https://www.maths.usyd.edu.au/u/uri/my_papers/2020_multicomp_RECOMB.pdf], yet with a large number of decoy databases instead of a small number of them. Of course, the authors' computationally efficient procedure using fewer decoy databases is preferable for daily use, but when it comes to evaluate an FDR control tool, resorting a single time on a larger number of decoy databases would make sense. More precisely, I think CRT could be valuable, as it could guarantee that even very long peptide sequences (which are not as frequent as shorter ones) have at least one decent competitors among the various randomized shuffle sequences (which is not the case notably with thinner windows for precursor mass tolerance, see above). For instance, it could be possible to design a CRT where each target sequence is guaranteed to have its best-decoy counterpart selected from a decoy subset of controlled size?

As the theoretical connections between these four approaches (BH, CRT, etc.) are strong, it would be even nicer to have many of them. I really believe the amount of work required to do so is not that important in comparison to what has already been achieved by the authors. Yet it would nicely fit with the discussions about the risk of circular reasoning while making the empirical results rock-solid.

To conclude, if the authors accept to (1) provide a more nuanced and formal context about the validity of the assumptions underlying entrapment and to (2) use orthogonal validation methods to exclude any risk of circular demonstration, I'll be honored to endorse their manuscript.

Thomas Burger

Reviewer #2:

Remarks to the Author:

In this manuscript, the authors present a critical evaluation of the existing entrapment-based protocols for validation of FDR estimates and propose an improved way of estimating FDP using a paired entrapment method. Results of the conducted computational experiments show that most of the evaluated DDA frameworks can control FDR at the peptide level, while the DIA approaches fail to do so at both peptide and protein levels.

While I agree that the issue of FDR control for DIA data in proteomics has become a major concern in the field, and this paper is timely and should be of interest to the proteomics community, the subject matter seems to be too technical in nature and specialized to warrant publication in Nature Methods, which has a much broader readership. I also found that the conceptual advance in using paired entrapment for FDR estimation to be incremental at best. The message this paper is expected to send to the community is definitely important, but in my view one would expect to see true conceptual or methodological advances in a Nature Methods paper.

Below are some comments and suggestions I have to improve the manuscript, whether or not it is eventually published in Nature Methods or some other more specialized journals.

1. I think the authors failed to address the criticism that using entrapment (shuffled sequences in this method) to validate FDR estimates by decoys (also shuffled sequences, just a different randomization) amounts to circular reasoning. Or put more specifically, the authors should explain why the "FDP" (probably wrong term, see Point 2 below) calculated from entrapment is not just another FDR estimate, no different from the one obtained from decoys. It is fine to suggest that having two "independent" FDR estimates to compare against each other would be helpful as a consistency check. (Now, even the "independence" part can be disputed because the two methods are so similar. The assumptions that shuffled decoys depends on are exactly the same ones that shuffled entrapment depends on.) But it is very different from saying that the entrapment-derived "FDP" is the ground truth and can be used to validate the decoy-derived "FDR" reported by the tools.

Further, the authors claim that their entrapment method can be used as long as the FDR estimation method does not use decoys in the same way as the entrapment method. The authors made a point that compromised FDR control in Percolator stems from “indirectly peeking at the target/decoy label” and as long as the post-processor has no knowledge of the entrapment sequences, then the entrapment matches can be reliably used for FDR control even if both entrapment and decoy sequences are shuffled in the same manner. However, I don’t find this too convincing. One of the issues with shuffled decoys, as it is well argued by many others, is that they do not fully capture some inherent patterns (e.g. homology, or some amino acids are more likely to be found next to each other than by random chance, etc.) in the target (real) sequences. The entrapment generated in the same manner will suffer from the same problem, and that was why the original entrapment idea uses real sequences from a foreign species. I grant that perhaps this approach can help to detect overfitting in a model like Percolator, but the entrapment-estimated FDR may still be wrong in the same way that decoy-estimated FDR (without using a machine learning-based post-processing tool like Percolator or the DIA tools) is wrong. Maybe a better way to sell the work and state their conclusion is to highlight the danger of overfitting in these machine learning models. To me, this is both more conservative (not claiming something one is not sure about), but also more forceful because it gets to the heart of the problem.

2. There are some semantic issues surrounding the use of the term false discovery proportion (FDP). Phrases such as “empirical FDP” and “the random nature of FDP” confuse me. I thought that the FDP is a true parameter of any given dataset, and is necessarily empirical. The “error rate” is an estimate of this parameter. The authors seem to have it backward when they state that the FDP is an “empirical error rate.” Even if one creates artificial datasets by randomization in a computational experiment for which the labels are known, each dataset has its own FDP value. Does the author imply that this “empirical” FDP is a realization of some underlying random variable? If so, what does this random variable mean conceptually? Or, perhaps what the authors call “FDP” is actually the estimate of the “empirical” FDP by their entrapment method. If so, how it is different from FDR? In Figure 2, they also used “estimated FDP” and “estimated FDR” interchangeably, which suggests the confusion is not mine only. It would be quite critical to define all these terms precisely up front so as not to confuse or mislead the reader.

3. I believe not putting more emphasis on DIA analysis is a missed opportunity. As the authors already stated, FDR control in DDA is already well-established, and thus their analyses just demonstrate that the paired variant of the entrapment method works correctly, in contrast to other entrapment variants. This can perhaps be placed in the Supplementary. On the other hand, the DIA section is merely one section at the end, with most of the analysis relegated to the Supplementary. However, problems with FDR control in DIA is exactly what we are most concerned about, and the most useful message of this paper. Perhaps the authors can say a little bit more about the spectrum-centric identification method used in DDA versus the peptide-centric identification used in DIA, and how that distinction affects the theory and practice of FDR control? To me those are apples and oranges, and yet we still define false discovery the same way, not to mention use the same FDR control method for both. Would that not be more interesting? The paper is far more impactful if the authors made the DIA part the main dish. The paired entrapment method can be presented as the “methodology” and the discussion about the incorrect use of entrapment by the community can be presented as literature review.

Some minor suggestions:

4. I would strongly suggest that the authors prioritize using more precise language to advance their arguments wherever possible. There are too many subjective or vague statements which clearly need elaboration and support. Perhaps I am overly sensitive, but if the goal of the paper is to challenge current understanding, the authors should try to make their writing as water-tight as possible.

5. For the estimated FDP plots, I suggest the authors provide a confidence band on top of the averaged FDP values (assuming they come from analysis of multiple entrapment sets). Then, we may get a better idea whether some methods really over- or underestimate the FDP, or the deviations we see are just due to the random nature of the estimation process. For example, in Figure 3, it is hard to judge whether the paired method is indeed more conservative than the combined method universally, or it is only true for the one dataset tested, since the curves are so close together. Not to mention that the authors tried to claim that for Tide+Percolator+RESET and Sage the paired method is conservative while the combined method is not. It may be true for this one particular test, but it is not clear the trend is general. As the authors suggested themselves, “FDR control should be universal; consequently, a valid FDR control procedure should achieve scenario (1) for any reasonably large dataset.” Perhaps they should apply the same standard to their own method.

6. Why is there no protein-level analysis for the DDA data, to mirror that for DIA data?

Reviewer #4:

Remarks to the Author:

A. Summary of the key results: The paper analyzes three entrapment based approaches for validating FDR error control approaches. It describes their pitfalls and proposes a novel approach based on an entrapment database of paired peptides and demonstrates its theoretical and empirical advantages. Using the current and novel validation approaches, it further demonstrates how most DIA search tools control FDR at a peptide level. However, similar conclusions could not be drawn for the DDA tools. And the issue becomes much worse for protein level FDR control.

B. Originality and significance: A novel approach entrapment based FDR validation approach is proposed

C. Data & methodology: approach is valid

D. Appropriate use of statistics and treatment of uncertainties: proofs are not rigorous, but intuitively the results should hold true. A result on sample entrapment approach as an overestimate is questionable. See detailed comments below.

E. Conclusions: The experimental results are reliable and valid. Though I'd like to more details in Figure 2a and b. See comments below.

F. Suggested improvements: see detailed comments below below

G. References: appropriate references are provided

H. Clarity and context: Needs improvement. See detailed comments below.

Overall comments: The paper has missing details at multiple places. The proofs and definitions are not rigorous. Though the most important results seem to be correct. The writing should be improved by adding missing details and making it paper easy to comprehend.

Weaknesses:

1) S1, S2, S5: The arguments in the proof seem to be made for peptide-level FDR only. For spectrum-level FDR control each spectrum would only have one correct peptide in the target database. All PSMs in T_p cannot be considered as a correct match for a given spectra. The authors should generalize the proof to that setting or explicitly state the limitations of their proof. In general, it seems that this work is not on PSM-level FDR. The authors should add statements in the text at multiple places that conveys the scope of this work clearly, even to a casual reader.

2) S1: The proof is not rigorous, could be vastly simplified and made more clear. It is not clear if the proof is about overestimating FDR or FDP. The expectations in the proof seem to convey a result on FDR. However, in that case the expectation should be taken on the ratio in Eq 1, rather than the numerator only. If the result is about FDP, a fixed quantity for a given experiment, it is not clear on what probability space the expectation is taken over. The space of the entrapment peptide database? The authors should define the true FDP to begin with. Shouldn't N_E be equal to V_E ? If so, please make that explicit. Please define V_X as false discoveries in database X, so that it generalizes to V_E, V_T, \dots , rather than defining $V_E + V_T$. Splitting the E into E_p and E_m looks like a convoluted way of proving a simple result. Technically, it would also require a way to handle randomness in the splitting. A simpler argument could be laid out based on the proportion of incorrect target peptides, say α . And $(\alpha/r)N_E$ being an unbiased estimator of the expected value of $V_{\{T_m\}}$, where the expectation is taken over multiple entrapment peptides database.

3) S2: The argument for sampling approach being an overestimate in the given example is based on reasoning that the true FDP is 0. However, that is only true when considering discoveries in the target. When considering the false discoveries in target + entrapment the true FDP is not necessarily 0. It is $N_E/(N_E+N_T)$

4) For the paired estimation approach the authors should specify how the score for each peptide is computed from the PSM scores, since approach assumes that each peptide is assigned a score. Is it based on the PSM-and-peptide approach in Section 4.1.2. Are all experiments performed with this approach? If so, please make it more explicit in the main document. Would the approach be still valid if the other peptide scoring from [21] were used?

5) Typically, error estimates derived from individual mass-spec experiments, perhaps incorrectly, are referred to as estimated/empirical FDR in the literature. Authors use the term estimated/empirical FDP for it and estimated/empirical FDR as an average over empirical FDP. However, they also claim that most tools report empirical FDR, implying that the tools average over the empirical FDP. Is that correct? Authors should make their use of terminology clear and also comment on how it differs from standard terminology.

6) To my knowledge, precursor Level FDR is not a well known term in mass spec. The authors should define it.

7) Please add sentences on how the 3 approaches could be extended to protein FDR and precursor FDR and provide any missing details.

Other comments:

8) The authors might want to give more intuition about an ideal entrapment method. Perhaps something like "It should give a tight upper bound to the true FDR. Being an upper bound ensures that if the FDR from the tool is greater than entrapment based estimate, it is also greater than the true FDR and hence gives a conservative list of discoveries. A tight upper bound ensures that it can be used for validation even when the difference between the estimated FDR and True FDR is small."

9) The term conservative FDR estimate could be confusing as a casual reader might interpret it to be lower than the true FDR. The authors should make it explicit what conservative FDR means.

10) Why were the combined approach not showed in Figure 2b?

11) Why was the true FDP not shown if Figure 2a, assuming only ISB18 discoveries as true discoveries, similar to Fig 2b?

12) The entrapment based approach for FDR validation is very similar to Target decoy approach for FDR control. Both essentially rely on the same assumption. If the Target decoy based FDR control is performed correctly by a tool, does

entrapment based validation have any utility? The authors might want to comment on this in the paper to further elaborate on the utility of entrapment based approaches.

Although we cannot publish your paper, it may be appropriate for another journal in the Nature Portfolio. If you wish to explore the journals and transfer your manuscript please use our manuscript transfer portal. You will not have to re-supply manuscript metadata and files, unless you wish to make modifications. For more information, please see our [manuscript transfer FAQ](http://www.nature.com/authors/author_resources/transfer_manuscripts.html?WT.mc_id=EMI_NPG_1511_AUTHORTRANSF&WT.ec_id=AUTHOR) page.

** For Nature Portfolio general information and news for authors, see <http://npg.nature.com/authors>.

Version 1:

Decision Letter:

5th Nov 2024

Dear Uri,

Thank you for your letter asking us to reconsider our decision on your Article, "Assessment of false discovery rate control in tandem mass spectrometry analysis using entrapment". After careful consideration we have decided that we are willing to consider a version of your manuscript that is revised in accordance with the plan proposed in your appeal letter.

- * include a point-by-point response to our referees and to any editorial suggestions
- * please underline/highlight any additions to the text or areas with other significant changes to facilitate review of the revised manuscript
- * address the points listed described below to conform to our open science requirements
- * ensure it complies with our general format requirements as set out in our guide to authors at www.nature.com/naturemethods
- * resubmit all the necessary files electronically by using the link below to access your home page

Link Redacted

We hope to receive your revised paper within 3-4 weeks. If you cannot send it within this time, please let us know. In this event, we will still be happy to reconsider your paper at a later date so long as nothing similar has been accepted for publication at Nature Methods or published elsewhere.

OPEN SCIENCE REQUIREMENTS

REPORTING SUMMARY AND EDITORIAL POLICY CHECKLISTS

When revising your manuscript, please submit reporting summary and editorial policy checklists.

DATA AVAILABILITY

CODE AVAILABILITY

Please include a "Code Availability" subsection in the Online Methods which details how your custom code is made available. Only in rare cases (where code is not central to the main conclusions of the paper) is the statement "available upon request" allowed (and reasons should be specified).

MATERIALS AVAILABILITY

SUPPLEMENTARY PROTOCOL

To help facilitate reproducibility and uptake of your method, we ask you to prepare a step-by-step Supplementary Protocol for the method described in this paper. We [encourage authors to share their step-by-step experimental protocols](https://www.nature.com/nature-research/editorial-policies/reporting-standards#protocols) on a protocol sharing platform of their choice and report the protocol DOI in the reference list. Nature Portfolio's protocols.io is a free-to-use and open resource for protocols; protocols deposited onto protocols.io are citable and can be linked from the published article. More details can found at [protocols.io](https://www.protocols.io/help/publish-articles).

ORCID

Sincerely,
Arunima

Arunima Singh, Ph.D.
Senior Editor
Nature Methods

Version 2:

Decision Letter:

24th Dec 2024

Dear Dr. Keich,

Your Article, "Assessment of false discovery rate control in tandem mass spectrometry analysis using entrapment", has now been seen by 2 reviewers. As you will see from their comments below, although the reviewers find your work of considerable potential interest, they have raised a few more concerns. We are interested in the possibility of publishing your paper in Nature Methods, but would like to consider your response to these concerns before we reach a final decision on publication.

We therefore invite you to revise your manuscript to address these concerns. I realize that many of these concerns pertain to data/method presentation, but given the technical nature of the paper we think it would be important to have the reviewers see the revised paper once more before accepting it for publication.

Link Redacted

We hope to receive your revised paper within 4-5 weeks. If you cannot send it within this time, please let us know. In this event, we will still be happy to reconsider your paper at a later date so long as nothing similar has been accepted for publication at Nature Methods or published elsewhere.

OPEN SCIENCE REQUIREMENTS

REPORTING SUMMARY AND EDITORIAL POLICY CHECKLISTS

EXTENDED DATA FIGURES

When re-submitting your manuscript, please ensure that any supplementary figures and tables that are crucial to the manuscript's conclusions are converted into Extended Data figures and tables to increase visibility of these data. Extended

Data figures and tables are online-only (present in the online PDF and full-text HTML versions of the paper), peer-reviewed display items that provide essential background to the article but are not included in the main article due to space constraints. A maximum of ten Extended Data display items (figures and tables) is permitted.

DATA AVAILABILITY

All novel DNA and RNA sequencing data, protein sequences, genetic polymorphisms, linked genotype and phenotype data, gene expression data, macromolecular structures, and proteomics data must be deposited in a publicly accessible database, and accession codes and associated hyperlinks must be provided in the "Data Availability" section.

CODE AVAILABILITY

Please include a "Code Availability" subsection in the Online Methods which details how your custom code is made available. Only in rare cases (where code is not central to the main conclusions of the paper) is the statement "available upon request" allowed (and reasons should be specified).

MATERIALS AVAILABILITY

SUPPLEMENTARY PROTOCOL

To help facilitate reproducibility and uptake of your method, we ask you to prepare a step-by-step Supplementary Protocol for the method described in this paper. We [encourage authors to share their step-by-step experimental protocols](https://www.nature.com/nature-research/editorial-policies/reporting-standards#protocols) on a protocol sharing platform of their choice and report the protocol DOI in the reference list. Nature Portfolio's protocols.io is a free-to-use and open resource for protocols; protocols deposited onto protocols.io are citable and can be linked from the published article. More details can found at [protocols.io](https://www.protocols.io/help/publish-articles).

ORCID

Nature Methods is committed to improving transparency in authorship. As part of our efforts in this direction, we are now requesting that all authors identified as 'corresponding author' on published papers create and link their Open Researcher and

Contributor Identifier (ORCID) with their account on the Manuscript Tracking System (MTS), prior to acceptance. This applies to primary research papers only. ORCID helps the scientific community achieve unambiguous attribution of all scholarly contributions. You can create and link your ORCID from the home page of the MTS by clicking on 'Modify my Springer Nature account'. For more information please visit www.springernature.com/orcid.

Sincerely,
Arunima

Arunima Singh, Ph.D.
Senior Editor
Nature Methods

Reviewers' Comments:

Reviewer #1 (Remarks to the Author):

The authors' revisions have well improved the manuscript. In addition to account for my comments, the authors made their points sharper while easier to read. I am still a supporter of this submission, but considering the journal reputation and how I expect this article to impact the community, I take the opportunity to propose another round of improvements:

1- Even if the revised manuscript is already improved with this respect, it is possible to be even clearer about the entrapment assumptions. Those are referred both in the introduction and conclusions, but they only fully appear in the supp mat, and there is no direct reference in the introduction and conclusions to the corresponding supp mat. I believe it would be nice to add to the main manuscript a synthetic table summarizing for each type of entrapment (combined, lower bound, paired/k-matched and if possible even the sample one, to highlight its lack of foundations) and each level (protein, peptide and even PSM, to highlight the associated lack of FDR control), the list of assumptions (under mathematical forms like eq. S1, Ass. 2, etc., but also the associated explanations at peptide/protein levels like in supp mat p 4 end of first paragraph) and discrepancies wrt to TDC (independence of false discoveries). Such summary would be helpful to a reader that has only rapidly screened the methods as it would give an overview of the limitations, assumptions and the technical barriers to proposing new validation methods.

2- Following the same logic, the authors refer a couple of times to the entrapment query protocol which Madej and Lam warned against. I understand their general setting is not incompatible with entrapment query, but the way it is phrased looks like the authors would like to re-open the door Madej and Lam tentatively closed. In my opinion, both views are however not incompatible: Madej and Lam warned about issues with the previous implementations, while the authors explain it could be implemented correctly in the future. A nice way to reconcile this would be to underline that although theoretically valid, implementing entrapment query protocols that provably fulfill the assumptions summarized in the table resulting from my previous comment is challenging and remains an unaddressed question.

3- In the previous round, I was partly unsatisfied with my lack of advice about a better display of the practical consequences of Table 1 results (essentially, that DIA is considered more sensitive than DDA, especially in SC but also in bulk analyses, partly because the FDR is not properly controlled; a point also raised by another reviewer) but in between, I got an idea: So far, the authors focused on comparing the claimed FDR control and the one obtained with FDRBench (First paragraph of p.10 and Supp mat S5). In themselves, the figures are striking, at least for a biostatistician, but my general feeling is that many wet-lab researchers would say "ok but shifting an FDR threshold from 1% to 2% is not a big deal, we essentially need a community standard, I am not sure it will change my understanding of the sample". Conversely, if the authors pinpoint that those FDR shifts lead to an artificial increment of X% of identified peptides or proteins, then things become concrete as any researcher comparing DDA and DIA will be able to evaluate which part of the DIA improved sensitivity is real, and which part was wrongly claimed as a result of an erroneous FDR control procedure. I find it would give ground to the assertions that the lack of statistical robustness of DIA tools have consequences as the authors claim in the introduction.

Minor:

- "Applying our entrapment procedures to DIA analysis tools we demonstrate that their FDR control at the peptide or precursor level can be questionable, and it is often questionable at the protein level." The result summary could be more nuanced: some FDR control are questionable, others are clearly invalid.
- when conditional probabilities involve cardinality, the equations tend to be less easy to read. Would the authors mind keeping "|" for conditioning and use an alternative notation (like "Card()" or "#[]" or whatever) for cardinality?

Reviewer #1 (Remarks on code availability):

code review would need longer time frames

Reviewer #2 (Remarks to the Author):

The revised manuscript has been much improved in terms of clarity and rigor. I would say that it is now up to the high standard of Nature Methods in those aspects, and my only reservation now is whether the focus and the scope is suitable for this journal.

I would not object to this paper being published in its current form. However, I still believe that a heavier emphasis on DIA will make the paper more impactful, as the unresolved problem of error control in DIA is perhaps the biggest issue in our field right now. It would have been nice to dig a bit deeper and drive the point home, and organize the paper's overall message around its most important finding. Challenging the accuracy of FDR estimates of those widely used DIA tools is the big deal here. But where is the word "DIA" in the paper title?

As it is now, the readers of this paper would spend most of the time reading about the various variants of entrapment-based FDR estimation/validation methods, which, let's be honest, is just too technical for the broader audience of proteomics practitioners. I am afraid that the extremely long setting of the stage will turn off most readers before it gets to the key message of the paper. If the paper were all about comparing different ways of calculating FDP from entrapment searches, and concluding that they mostly work for DDA, it should not be published in Nature Methods. (I imagine that even the authors would agree with me on that.) In other words, the DIA part is the main reason why we are all excited about the work and feel that it deserves a bigger platform.

Version 3:

Decision Letter:

Our ref: NMETH-A56676C

27th Jan 2025

Dear Uri,

Thank you for submitting your revised manuscript "Assessment of false discovery rate control in tandem mass spectrometry analysis using entrapment" (NMETH-A56676C). It has now been seen by the original referees and their comments are below. The reviewers find that the paper has improved in revision, and therefore we'll be happy in principle to publish it in Nature Methods, pending minor revisions to satisfy the referees' final requests and to comply with our editorial and formatting guidelines.

TRANSPARENT PEER REVIEW

ORCID

Sincerely,
Arunima

Arunima Singh, Ph.D.
Senior Editor
Nature Methods

Reviewer #1 (Remarks to the Author):

The newest version of the paper suits me fine, I think it can be published as is, and I expect it will positively influence the future literature.

Reviewer #1 (Remarks on code availability):

10 days to review the paper plus the code is too short

Reviewer #2 (Remarks to the Author):

Although I would still prefer that the paper takes a stronger stance about DIA error control and reduces the emphasis on the technical details of entrapment analysis, I respect the authors' decision to keep it in the current form. I have no further comments about the latest revisions.

Version 4:

Decision Letter:

24th Apr 2025

Dear Uri,

I am pleased to inform you that your Article, "Assessment of false discovery rate control in tandem mass spectrometry analysis using entrapment", has now been accepted for publication in Nature Methods. The received and accepted dates will be June 3, 2024 and April 24, 2025. This note is intended to let you know what to expect from us over the next month or so, and to let you know where to address any further questions.

Over the next few weeks, your paper will be copyedited to ensure that it conforms to Nature Methods style. Once your paper is typeset, you will receive an email with a link to choose the appropriate publishing options for your paper and our Author Services team will be in touch regarding any additional information that may be required. It is extremely important that you let us know now whether you will be difficult to contact over the next month. If this is the case, we ask that you send us the contact information (email, phone and fax) of someone who will be able to check the proofs and deal with any last-minute problems.

If you are active on Twitter/X or Bluesky, please e-mail me your and your coauthors' handles so that we may tag you when the paper is published.

Best regards,
Arunima

Arunima Singh, Ph.D.
Senior Editor
Nature Methods

** Visit the Springer Nature Editorial and Publishing website at http://editorial-jobs.springernature.com?utm_source=ejP_NMeth_email&utm_medium=ejP_NMeth_email&utm_campaign=ejp_Nmeth for more information about our career opportunities. If you have any questions please click [here](mailto:editorial.publishing.jobs@springernature.com).**

October 5, 2024

Dear Dr. Singh:

We thank you and the reviewers for your kind consideration of our manuscript. Below, we address each of the points raised by the three reviewers and describe the changes we have made to the manuscript. In what follows, the reviewer's comments are shown in black type, interleaved with our responses (in blue) and, where appropriate, the modified text (in red).

Thank you very much for your consideration.

Best regards,

Uri Keich, Associate Professor
School of Mathematics and Statistics
University of Sydney

William Noble, Professor
Department of Genome Sciences
University of Washington

Reviewer 1

In this paper, Wen et al. focus on the correct way to validate FDR control tools applied to the peptide identification problem in proteomics, using benchmarking methods build on the concept of “entrapment database”. The contributions are multiple and important:

We thank the reviewer for their positive assessment of our manuscript. In the revised version we tried to clarify that our analyses and proposed tools apply more generally than in the peptide level analysis (which we often use as a concrete example).

1- They provide actionable guidelines intended to the computational proteomic experts, to avoid the entrapment concept is used incorrectly, or implemented into a flawed procedure. General comment: In my opinion, this contribution is important to very important, considering the current lack of rigor in the field, but it addresses a niche readership (essentially proteomic software developers).

Entrapment is used more widely than just proteomic software developers because it is often used in comparative and benchmarking analyses. Indeed, some of the papers we mention in Table 1 are such.

More critically, this contribution is not presented in a sufficiently formal way, so that it may miss its goal. Note that here, “insufficiently formal” does not refer to the statistical correctness of the ideas exposed (which from my viewpoint is excellent), but to how it should be concretely applied by non-experts (points of vigilance, limitations, precise itemization of the recipe, risk of “circular reasoning” as reported by Madej and Lam [25], etc.). Therefore, I believe improving the manuscript with this respect should essentially require adjusting the presentation (see details below).

We agree with the reviewer that our initial exposition could have significantly benefitted from a more formal approach, and we revised the manuscript accordingly as detailed below.

2- They propose a new (tighter) upper bound on the FDR (plus the associated software). General comment: The proposal is theoretically sounded, while carefully evaluated on the empirical side, notably using a clever double-entrapment strategy. Likewise, providing software to improve future evaluation is valuable. However, there is one blind spot of importance, which relates to the same issue as the point above, namely the possible limits of the evaluation owing to the risk of circular reasoning (broadly speaking, “I assume the equal chance assumption (ECA) holds between mismatches and matches onto shuffled sequences. Therefore TDC-based FDR should be correct, but to empirically demonstrate it, I will rely on an entrapment approach which requires the exact same assumption”, see [25]). I do not doubt on the correctness of the new bound (the theoretical support is solid) and I am positive about this contribution, but I think an orthogonal empirical validation is nevertheless necessary.

We addressed this request in three ways: (1) explicitly laying out the assumptions that our new estimate relies on and rigorously proving its validity when those assumptions hold, (2) more explicitly addressing the circular reason argument in the Discussion section, and (3) applying the entrapment estimation method to procedures that do not use TDC to control the FDR (as requested by the reviewer). More details on these points are provided below. Additionally, as we point out in the revised discussion, we provide multiple figures showing the agreement between using shuffled and foreign entrapment sequences.

3- They expose the lack of rigor about how FDR control was validated in the currently used tools, specially DIA ones. General comment: The (dramatic) conclusion is very timely, extremely interesting, and calls for a mass reaction in the proteomics community, considering the recent hype about using DIA approaches, notably for single-cell proteomics. Based on the manuscript, it is tempting to assume that the main reason why DIA tools are preferred for SC proteomics is because a less strict FDR control makes it possible to have longer lists of identified peptides, regardless the DDA vs. DIA data quality. As is, this contribution should have a long-standing echo in the community. I am totally supporting the authors’ analysis, and I wonder about making it more explicit in the title or the abstract of the paper (BTW, I found the abstract too long

and too technical and I assume a catchier line would better support the paper... Unfortunately, I have no concrete proposal to help the authors in this direction yet).

Overall, I support this article and I would be pleased to see it published in Nature Methods in the end. However, I am partially unsatisfied with some aspects of the current version, in relation to points 1 and 2 above. In summary, this paper has the potential to close a decade-old wandering in the computational proteomics community with respect to the correct use of entrapment sequences, and this must be acknowledged. However, this wandering originated in the fuzziness of the earlier papers about entrapment, as purposely highlighted by Madej and Lam [25] (BTW, there is a typo in the paper, Majed instead of Madej): the original lack of precise definitions and of hypotheses led the possibility to elaborate sophisticated entrapment-based procedure, more or less correct, and implicitly rooted on circular reasoning. To avoid perpetuating this, I think it is important the authors provide the clearest possible guidelines and accept discussing in more details the limitations of the approach. Concretely:

We thank the reviewer for their support and for their following constructive suggestions.

1- Provide a more formal and more precise definition of entrapment: - In [25], Madej and Lam distinguish different uses or types of entrapment. Does the authors agree on this classification? And which type of entrapment they focus on? (e.g., as pointed in supp mat, the authors prefer shuffling over distant organism, but this should be stated formally and earlier, as to better delineate the scope). - Likewise the authors of [25] root the entrapment concept into publications earlier than [10]. This may not old depending on the type of entrapment approaches the authors focus one, but still, clarification is needed from the readership viewpoint, as to delineate which methods the proposed recommendations apply to.

We added a formal framework that models both types of experiments that Madej and Lam discuss (Supplementary Section S1, “An abstraction of the entrapment concept”). This model captures all the types of entrapment experiments the reviewer discusses, and the estimation methods we investigate are equally applicable to all as well, which we establish in new Supplementary Section S2.

As for the referenced papers, the earlier one cited by Madej and Lam references a supplement that is not available so we cannot figure out what exactly was done there. The later paper cited by them, like Madej and Lam’s own work, uses the sample estimation method which, as we argue here, is invalid. Regardless, we added the later paper, which at least provides the method’s details.

2- Provide a more formal definition of the hypotheses entrapment is based on and discuss whether decoys and entrapment sequences are exactly the same or not (hypotheses for correct generation, and how they should be used) as this distinction is at the core of a possible circular reasoning.

We added a new Supplementary Section S2 that contains clearly defined assumptions based on which we can rigorously prove the properties of the estimation methods. As we explain in that section, as well as in the Discussion, the validity of the assumptions depends on what we refer to as the expansion of the dataset (either through entrapment sequences or entrapment spectra). This paper focuses on the estimation component of the entrapment experiment, and we anchored it in rigorous mathematical theory. We touch on issues that are related to the expansion and explain why we chose to focus mostly on shuffled sequences and which assumptions we believe are reasonably expected to hold. That said, we believe the topic of expansion merits further research.

3- Provide a more formal description of the entrapment implementation (like with a pseudo-algorithm for instance?).

We added Supplementary Algorithms 1 and 2 that describe how we generate, respectively, shuffled and foreign entrapment peptides. As for the estimation methods, we believe they are well described both in the main paper in Equations (1)-(4) and in Supplementary Section S2.

4- I believe the authors overlook the circular reasoning highlighted by Madej et al. Notably they say: “We only partially agree with this statement. Specifically, we contend that this reservation is valid only if the

analysis tool's use of the decoy sequences is restricted to estimating the FDR using the same estimation method that the entrapment relies on." I agree the case highlighted is problematic. However, contrarily to what the authors claim, it is not the only possible case: Depending on how the shuffle sequences (be them entrapment ones or decoys) distribute (because of a flawed generation process or an artifact that affects both procedures) and how this impacts the competition against real sequences, the entrapment validation may be equally flawed, despite relying on a different mechanism. Here are 3 examples:

* Percolator: In addition to the issue pointed in [8], percolator can learn to distinguish shuffled sequences from target ones <https://doi.org/10.1021/acs.jproteome.8b00991>, leading to prefer the latter ones to the former ones (in other words, the ECA no longer holds), and thus to underestimating the FDR. If shuffled sequences are also used for the entrapment validation, the same problem will lead to hiding the underestimated FDR, as percolator will recognize the entrapment sequences as easily as the decoy ones.

* Proteogenomics: In <https://doi.org/10.1186/s13059-022-02701-2>, we have empirically demonstrated that the reduction of the target database size according to transcriptomic assays can break the ECA (briefly, because in the process, the best decoy score is lowered much more than the best target one, leading to an imbalanced competition). How entrapment would be affected by this (as it also requires the ECA to hold) is not clear. I do not think it is possible to make concrete conclusions without investigating this, and as the author say, the onus fall on the developer.

* We have shown the same with the reduction of the precursor mass tolerance windows <https://doi.org/10.1021/acs.analchem.0c00328>. We observed this phenomenon did not affect target-decoy without competition, and I personally believe (even though I have not been able to demonstrate it so far) that subsetting the databases (using the precursor windows or the transcriptomic data) leads to breaking the ECA. To support this intuition, consider the case of only 1 target sequence T falling in the windows: it will be selected as the best target regardless its score. If T is incorrect but almost correct (eg, only a couple of AA permutations) the probability that its shuffled version outscores T is much smaller than if several shuffled sequence would have fallen in the mass tolerance windows. These phenomena are not considered in [11] (notably Theorem 2), as the ECA is taken for granted whatever the TDC setting. However, considering the empirical evidence, it is legitimate to acknowledge the ECA may sometime not hold.

To conclude, any situation where the ECA may not hold any longer, be it because of how the data are processed, of how the FDR is controlled in the TDC procedure, or of how sequences are shuffled, can lead to flawed entrapment procedures. Finally, there are many situations beyond what the authors contend where the circular reasoning is a risk, and this should be better discussed and delineated, notably thanks to more formal presentations of the definitions, hypotheses, implementations, etc. (as commented above). This said, I acknowledge that in the many cases where the ECA holds, entrapment as presented by the authors is valid, so my point is not to minimize the nice results of the paper, but only to better delineate their validity domain.

We thank the reviewer for pointing out this deficiency in our arguments. We added to the revised Discussion section an extended discussion regarding the possible circular reasoning and how we believe we addressed it, including highlighting various experiments we did that used both foreign and shuffled experiments, thereby significantly alleviating the concern of circular reasoning.

5- The previous point explains why I found a blind spot in the empirical evaluation of the new tighter upper bound on the FDP. Considering competition against shuffle sequences are used both for the validation (entrapment) and the FDR control (target-decoy competition), circular reasoning cannot be excluded. The only way to overcome this is to propose an orthogonal validation, like for instance:

* Target-decoy without competition

* Benjamini-Hochberg procedure

* Empirical null as proposed in early papers by Nesvizhskii.

* Conditional randomization test: essentially the same idea the authors developed in https://www.maths.usyd.edu.au/u/uri/my_papers/2020_multicomp_RECMB.pdf, yet with a large number of decoy databases instead of a small number of them.

Of course, the authors' computationally efficient procedure using fewer decoy databases is preferable for daily use, but when it comes to evaluate an FDR control tool, resorting a single time on a larger number of decoy databases would make sense. More precisely, I think CRT could be valuable, as it could guarantee that even very long peptide sequences (which are not as frequent as shorter ones) have at least one decent competitors among the various randomized shuffle sequences (which is not the case notably with thinner windows for precursor mass tolerance, see above). For instance, it could be possible to design a CRT where each target sequence is guaranteed to have its best-decoy counterpart selected from a decoy subset of controlled size? As the theoretical connections between these four approaches (BH, CRT, etc.) are strong, it would be even nicer to have many of them. I really believe the amount of work required to do so is not that important in comparison to what has already been achieved by the authors. Yet it would nicely fit with the discussions about the risk of circular reasoning while making the empirical results rock-solid.

Following the reviewer's suggestion, we conducted a new experiment that falls under the category that the reviewer refers to as "Target-decoy without competition". As described at the end of Section 2.4, we applied the studied estimation methods to peptide-level analysis procedures that do not rely on TDC to control the FDR. Specifically, we analyzed two such procedures in the spirit of the PSM-level one described in [1]: each peptide was assigned an empirical p-value, derived from the aggregated decoy peptide scores. The results of this experiment, which was repeated using both shuffled and foreign entrapment sequences, are consistent with the ones where the analyzed procedure was TDC based. Notably, the entrapment experiments indicate that these no-competition procedures are less powerful than the TDC based one, which is consistent with the fact that they report fewer discovered peptides.

To conclude, if the authors accept to (1) provide a more nuanced and formal context about the validity of the assumptions underlying entrapment and to (2) use orthogonal validation methods to exclude any risk of circular demonstration, I'll be honored to endorse their manuscript.

We thank the reviewer for offering us this route, which we believe we successfully took.

Reviewer 2

In this manuscript, the authors present a critical evaluation of the existing entrapment-based protocols for validation of FDR estimates and propose an improved way of estimating FDP using a paired entrapment method. Results of the conducted computational experiments show that most of the evaluated DDA frameworks can control FDR at the peptide level, while the DIA approaches fail to do so at both peptide and protein levels.

While I agree that the issue of FDR control for DIA data in proteomics has become a major concern in the field, and this paper is timely and should be of interest to the proteomics community, the subject matter seems to be too technical in nature and specialized to warrant publication in Nature Methods, which has a much broader readership. I also found that the conceptual advance in using paired entrapment for FDR estimation to be incremental at best. The message this paper is expected to send to the community is definitely important, but in my view one would expect to see true conceptual or methodological advances in a Nature Methods paper.

Below are some comments and suggestions I have to improve the manuscript, whether or not it is eventually published in Nature Methods or some other more specialized journals.

Our literature review shows that there is much confusion in the field regarding the estimation step of an entrapment procedure. Indeed, Table 1 shows that even papers published in Nature Methods relied on false arguments in trying to establish FDR control. As such, we believe that this paper, which is the first to formally lay out how entrapment estimation should be done, does represent a major conceptual advance in the field.

1. I think the authors failed to address the criticism that using entrapment (shuffled sequences in this method) to validate FDR estimates by decoys (also shuffled sequences, just a different randomization) amounts to circular reasoning. Or put more specifically, the authors should explain why the “FDP” (probably wrong term, see Point 2 below) calculated from entrapment is not just another FDR estimate, no different from the one obtained from decoys. It is fine to suggest that having two “independent” FDR estimates to compare against each other would be helpful as a consistency check. (Now, even the “independence” part can be disputed because the two methods are so similar. The assumptions that shuffled decoys depends on are exactly the same ones that shuffled entrapment depends on.) But it is very different from saying that the entrapment-derived “FDP” is the ground truth and can be used to validate the decoy-derived “FDR” reported by the tools.

Further, the authors claim that their entrapment method can be used as long as the FDR estimation method does not use decoys in the same way as the entrapment method. The authors made a point that compromised FDR control in Percolator stems from “indirectly peeking at the target/decoy label” and as long as the post-processor has no knowledge of the entrapment sequences, then the entrapment matches can be reliably used for FDR control even if both entrapment and decoy sequences are shuffled in the same manner. However, I don’t find this too convincing. One of the issues with shuffled decoys, as it is well argued by many others, is that they do not fully capture some inherent patterns (e.g. homology, or some amino acids are more likely to be found next to each other than by random chance, etc.) in the target (real) sequences. The entrapment generated in the same manner will suffer from the same problem, and that was why the original entrapment idea uses real sequences from a foreign species. I grant that perhaps this approach can help to detect overfitting in a model like Percolator, but the entrapment-estimated FDR may still be wrong in the same way that decoy-estimated FDR (without using a machine learning-based post-processing tool like Percolator or the DIA tools) is wrong. Maybe a better way to sell the work and state their conclusion is to highlight the danger of overfitting in these machine learning models. To me, this is both more conservative (not claiming something one is not sure about), but also more forceful because it gets to the heart of the problem.

We thank the reviewer for pointing out the shortcomings of our initial argument. In response to these concerns, as well as similar concerns from Reviewer #1, we revised the Discussion section to provide a more detailed discussion regarding potential circular reasoning. This new text points out that the DIA

analysis relies mostly on the lower bound estimate which, as we explain, is not exposed to the risk of circular reasoning. We also highlight several lines of evidence we provide that alleviate the circular reasoning concern by showing that shuffled-based entrapment agree with foreign-based entrapment. That said, this paper focuses on developing a formal framework within which the estimation methods can be rigorously analyzed subject to the entrapment expansion (e.g., through construction of an entrapment database) satisfying the outlined assumptions. We believe that the in-depth study of entrapment expansion merits future research.

2. There are some semantic issues surrounding the use of the term false discovery proportion (FDP). Phrases such as “empirical FDP” and “the random nature of FDP” confuse me. I thought that the FDP is a true parameter of any given dataset, and is necessarily empirical. The “error rate” is an estimate of this parameter. The authors seem to have it backward when they state that the FDP is an “empirical error rate.” Even if one creates artificial datasets by randomization in a computational experiment for which the labels are known, each dataset has its own FDP value. Does the author imply that this “empirical” FDP is a realization of some underlying random variable? If so, what does this random variable mean conceptually? Or, perhaps what the authors call “FDP” is actually the estimate of the “empirical” FDP by their entrapment method. If so, how it is different from FDR? In Figure 2, they also used “estimated FDP” and “estimated FDR” interchangeably, which suggests the confusion is not mine only. It would be quite critical to define all these terms precisely up front so as not to confuse or mislead the reader.

We apologize for contributing to this confusion by occasionally using imprecise terminology. The correct statistical term for the proportion of false discovery in a single experiment is the FDP, which is indeed a random variable rather than a parameter. The FDR is the expected value, or the average of the FDP over all random aspects, and as such can be considered a parameter.

While entrapment experiments typically estimate the FDP, in some of our analyses we average that estimate over multiple decoy and entrapment sets; hence, we refer to this average as “estimated FDR.” We carefully went over the manuscript and fixed all the loosely used terms, including modifying the axis labels in Figure 2, as well as the caption of some of the supplementary figures, and clarifying our terminology at the very start of the revised introduction.

3. I believe not putting more emphasis on DIA analysis is a missed opportunity. As the authors already stated, FDR control in DDA is already well-established, and thus their analyses just demonstrate that the paired variant of the entrapment method works correctly, in contrast to other entrapment variants. This can perhaps be placed in the Supplementary. On the other hand, the DIA section is merely one section at the end, with most of the analysis relegated to the Supplementary. However, problems with FDR control in DIA is exactly what we are most concerned about, and the most useful message of this paper. Perhaps the authors can say a little bit more about the spectrum-centric identification method used in DDA versus the peptide-centric identification used in DIA, and how that distinction affects the theory and practice of FDR control? To me those are apples and oranges, and yet we still define false discovery the same way, not to mention use the same FDR control method for both. Would that not be more interesting? The paper is far more impactful if the authors made the DIA part the main dish. The paired entrapment method can be presented as the “methodology” and the discussion about the incorrect use of entrapment by the community can be presented as literature review.

We revised the presentation to put more emphasis on the DIA results. However, from our perspective, the major contribution is still the methodological one. Diving deeper into the reasons behind the differences between DIA and DDA results should be an interesting subject for future research.

Some minor suggestions:

4. I would strongly suggest that the authors prioritize using more precise language to advance their arguments wherever possible. There are too many subjective or vague statements which clearly need elaboration and support. Perhaps I am overly sensitive, but if the goal of the paper is to challenge current understanding, the authors should try to make their writing as water-tight as possible.

We carefully edited the manuscript and we believe that, together with the newly introduced formal frame-

work, we managed to remove most of the subjective and ambiguous statements.

5. For the estimated FDP plots, I suggest the authors provide a confidence band on top of the averaged FDP values (assuming they come from analysis of multiple entrapment sets). Then, we may get a better idea whether some methods really over- or underestimate the FDP, or the deviations we see are just due to the random nature of the estimation process. For example, in Figure 3, it is hard to judge whether the paired method is indeed more conservative than the combined method universally, or it is only true for the one dataset tested, since the curves are so close together. Not to mention that the authors tried to claim that for Tide+Percolator+RESET and Sage the paired method is conservative while the combined method is not. It may be true for this one particular test, but it is not clear the trend is general. As the authors suggested themselves, “FDR control should be universal; consequently, a valid FDR control procedure should achieve scenario (1) for any reasonably large dataset.” Perhaps they should apply the same standard to their own method.

The lower and upper bound nature of the estimates were argued theoretically and are demonstrated empirically in Figures 2 and 3. As per the reviewer’s suggestion, we added 95% coverage bands to Figure 2a. We also added the new Supplementary Figure S5 which provides the 95% coverage bands for the combined and paired estimated FDR in the case of Tide and Sage. Note that MS-GF+ was too slow and MSFragger requires a different application for each FDR threshold, so both were omitted here.

6. Why is there no protein-level analysis for the DDA data, to mirror that for DIA data?

We added protein-level analysis using MaxQuant. As the reviewer suggested, we mostly used the DDA data to demonstrate our claims regarding the nature of the considered estimates in a context that is well studied. Hence, we also restricted our DDA analysis to one controlled experiment (ISB18) and one typical experiment (HEK293).

Reviewer 4

A. Summary of the key results: The paper analyzes three entrapment based approaches for validating FDR error control approaches. It describes their pitfalls and proposes a novel approach based on an entrapment database of paired peptides and demonstrates its theoretical and empirical advantages. Using the current and novel validation approaches, it further demonstrates how most DIA search tools control FDR at a peptide level. However, similar conclusions could not be drawn for the DDA tools. And the issue becomes much worse for protein level FDR control.

B. Originality and significance: A novel approach entrapment based FDR validation approach is proposed

C. Data & methodology: approach is valid

D. Appropriate use of statistics and treatment of uncertainties: proofs are not rigorous, but intuitively the results should hold true. A result on sample entrapment approach as an overestimate is questionable. See detailed comments below.

E. Conclusions: The experimental results are reliable and valid. Though I'd like to more details in Figure 2a and b. See comments below.

F. Suggested improvements: see detailed comments below below

G. References: appropriate references are provided

H. Clarity and context: Needs improvement. See detailed comments below.

Overall comments: The paper has missing details at multiple places. The proofs and definitions are not rigorous. Though the most important results seem to be correct. The writing should be improved by adding missing details and making it paper easy to comprehend.

We thank the reviewer for their constructive comments which we believe we addressed as detailed below.

Weaknesses:

1) S1,S2, S5: The arguments in the proof seem to be made for peptide-level FDR only. For spectrum-level FDR control each spectrum would only have one correct peptide in the target database. All PSMs in T_p cannot be considered as a correct match for a given spectra. The authors should generalize the proof to that setting or explicitly state the limitations of their proof. In general, it seems that this work is not on PSM-level FDR. The authors should add statements in the text at multiple places that conveys the scope of this work clearly, even to a casual reader.

We and others have pointed out that it is impossible to control the FDR at the PSM level in the usual framework of TDC, because of the potential lack of independence among the incorrect PSMs [2, 3]. That said, our entrapment setup can in principle be applied to rigorously gauge the FDP at the PSM level because our analysis of the estimators show that they do not require that problematic independence assumption. In addition, we clarified in the revision that our analysis applies to all three levels: PSM, peptide and protein.

2) S1: The proof is not rigorous, could be vastly simplified and made more clear. It is not clear if the proof is about overestimating FDR or FDP. The expectations in the proof seem to convey a result on FDR. However, in that case the expectation should be taken on the ratio in Eq 1, rather than the numerator only. If the result is about FDP, a fixed quantity for a given experiment, it is not clear on what probability space the expectation is taken over. The space of the entrapment peptide database? The authors should define the true FDP to begin with. Shouldn't N_E be equal to V_E ? If so, please make that explicit. Please define V_X as false discoveries in database X, so that it generalizes to V_E, V_T, \dots , rather than defining $V_E + V_T$. Splitting the E into E_p and E_m looks like a convoluted way of proving a simple result. Technically, it would also require a way to handle randomness in the splitting. A simpler argument could be laid out based on the proportion of incorrect target peptides, say α . And $(\alpha/r)N_E$ being an unbiased estimator of the expected value of V_{T_m} , where the expectation is taken over multiple entrapment peptides database.

We thank the reviewer for pointing out these deficiencies. In response, we completely rewrote these sections to include a new formal definition of an entrapment experiment and the explicit assumptions that are required

in order to prove the claimed properties of the various estimation methods that we study. The mathematical statements themselves were also overhauled and are now rigorously proved rather than argued intuitively.

3) S2: The argument for sampling approach being an overestimate in the given example is based on reasoning that the true FDP is 0. However, that is only true when considering discoveries in the target. When considering the false discoveries in target + entrapment the true FDP is not necessarily 0. It is $N_E/(N_E+N_T)$

We thank the reviewer for catching the error with our previous argument. We believe we fixed the problem in the revised version (highlighted in red in Supplementary Section S3).

4) For the paired estimation approach the authors should specify how the score for each peptide is computed from the PSM scores, since approach assumes that each peptide is assigned a score. Is it based on the PSM-and-peptide approach in Section 4.1.2. Are all experiments performed with this approach? If so, please make it more explicit in the main document. Would the approach be still valid if the other peptide scoring from [21] were used?

In our new formulation we clarify the general conditions under which the paired method is applicable, and we point out that it is not restricted to the PSM-and-peptide procedure.

5) Typically, error estimates derived from individual mass-spec experiments, perhaps incorrectly, are referred to as estimated/empirical FDR in the literature. Authors use the term estimated/empirical FDP for it and estimated/empirical FDR as an average over empirical FDP. However, they also claim that most tools report empirical FDR, implying that the tools average over the empirical FDP. Is that correct? Authors should make their use of terminology clear and also comment on how it differs from standard terminology.

We apologize for contributing to this confusion. The correct statistical term for the proportion of false discovery in a single experiment is the FDP, and the FDR is the expected value or the average of the FDP over all random aspects. We tried to clarify this terminology at the very start of the introduction in our revision. We also fixed the incorrect references to “empirical FDR” in some of the supplementary figures that should have referred to “estimated FDP” instead.

6) To my knowledge, precursor Level FDR is not a well known term in mass spec. The authors should define it.

We added to Section 2.5 and Supplementary Section S1 brief comments explaining this term.

7) Please add sentences on how the 3 approaches could be extended to protein FDR and precursor FDR and provide any missing details.

The new presentation clarifies that the estimation methods are applicable to all levels of analysis.

Other comments: 8) The authors might want to give more intuition about an ideal entrapment method. Perhaps something like “It should give a tight upper bound to the true FDR. Being an upper bound ensures that if the FDR from the tool is greater than entrapment based estimate, it is also greater than the true FDR and hence gives a conservative list of discoveries. A tight upper bound ensures that it can be used for validation even when the difference between the estimated FDR and True FDR is small.”

We added a discussion along these line to the first paragraph of Section 2.2.

9) The term conservative FDR estimate could be confusing as a casual reader might interpret it to be lower than the true FDR. The authors should make it explicit what conservative FDR means.

We revised the language accordingly, either removing the reference to “conservative” or adding a clarifying context.

10) Why were the combined approach not showed in Figure 2b?

The revised figure includes all 4 estimation methods, and the discussion of the figure in Section 2.3 was updated accordingly.

11) Why was the true FDP not shown in Figure 2a, assuming only ISB18 discoveries as true discoveries, similar to Fig 2b?

In this case, the true FDP would coincide with the lower bound. However, note that there is a 1:1 correspondence here between the entrapment and the ISB18, so the assumption that every reported ISB18 peptide is correct is unwarranted. In contrast, in 2b the “original target” is made of the ISB18 and the much larger castor proteome, so it is reasonable to assume that any reported ISB18 peptide is correct.

12) The entrapment based approach for FDR validation is very similar to Target decoy approach for FDR control. Both essentially rely on the same assumption. If the Target decoy based FDR control is performed correctly by a tool, does entrapment based validation have any utility? The authors might want to comment on this in the paper to further elaborate on the utility of entrapment based approaches.

The reviewer is probably alluding to the same circular reasoning concern that the other two reviewers raised and which we believed we addressed as explained above.

References

- [1] L. Käll, J. D. Storey, M. J. MacCoss, and W. S. Noble. Assigning significance to peptides identified by tandem mass spectrometry using decoy databases. *Journal of Proteome Research*, 7(1):29–34, 2008.
- [2] K. He, Y. Fu, W.-F. Zeng, L. Luo, H. Chi, C. Liu, L.-Y. Qing, R.-X. Sun, and S.-M. He. A theoretical foundation of the target-decoy search strategy for false discovery rate control in proteomics. *arXiv*, 2015. <https://arxiv.org/abs/1501.00537>.
- [3] A. Lin, T. Short, W. S. Noble, and U. Keich. Improving peptide-level mass spectrometry analysis via double competition. *Journal of Proteome Research*, 21(10):2412–2420, 2022.

January 7, 2025

Dear Dr. Singh:

We thank you and the reviewers for your kind consideration of our manuscript. Below, we address each of the points raised by the two reviewers and describe the changes we have made to the manuscript. In what follows, the reviewer's comments are shown in black type, interleaved with our responses (in blue) and, where appropriate, the modified text (in red).

Thank you very much for your consideration.

Best regards,

Uri Keich, Associate Professor
School of Mathematics and Statistics
University of Sydney

William Noble, Professor
Department of Genome Sciences
University of Washington

Reviewer 1

The authors’ revisions have well improved the manuscript. In addition to account for my comments, the authors made their points sharper while easier to read. I am still a supporter of this submission, but considering the journal reputation and how I expect this article to impact the community, I take the opportunity to propose another round of improvements:

We thank the reviewer for their support and for their following constructive suggestions.

1- Even if the revised manuscript is already improved with this respect, it is possible to be even clearer about the entrapment assumptions. Those are referred both in the introduction and conclusions, but they only fully appear in the supp mat, and there is no direct reference in the introduction and conclusions to the corresponding supp mat. I believe it would be nice to add to the main manuscript a synthetic table summarizing for each type of entrapment (combined, lower bound, paired/k-matched and if possible even the sample one, to highlight its lack of foundations) and each level (protein, peptide and even PSM, to highlight the associated lack of FDR control), the list of assumptions (under mathematical forms like eq. S1, Ass. 2, etc., but also the associated explanations at peptide/protein levels like in supp mat p 4 end of first paragraph) and discrepancies wrt to TDC (independence of false discoveries). Such summary would be helpful to a reader that has only rapidly screened the methods as it would give an overview of the limitations, assumptions and the technical barriers to proposing new validations methods.

We added the suggested table at the beginning of Supplementary Section S2 as well as a reference to it in the first paragraph of the discussion section. We chose not to include in the table the erroneous sample estimation methods because these do not fit our framework. The table is reproduced here as Table 1.

Name	Notation	Expression	Required Assumptions and Comments
Lower Bound	$\widehat{\text{FDP}}_{\mathcal{E} \cup \mathcal{O}}$	$\frac{N_{\mathcal{E}}}{N_{\mathcal{O}} + N_{\mathcal{E}}}$	Every entrapment discovery is a false one (Assumption 1, see also Assumptions 1a and 1b for common use cases).
Combined (average upper bound)	$\widehat{\text{FDP}}_{\mathcal{E} \cup \mathcal{O}}$	$\frac{N_{\mathcal{E}}(1+1/r)}{N_{\mathcal{O}} + N_{\mathcal{E}}}$	Given the number of discoveries K_{α} , any false discovery is at least r times more likely to be an entrapment than an original discovery (Assumption 2). See also Assumption 2a and the discussion following it for common use cases.
Matched (average upper bound)	$\widehat{\text{FDP}}_{\mathcal{E} \cup \mathcal{O}}^{*k}$	$\frac{N_{\mathcal{E}} + \sum_{l=1}^{k+1} l \cdot N_l}{N_{\mathcal{O}} + N_{\mathcal{E}}}$	Every potential original discovery is matched with k potential entrapment discoveries. Roughly, when ordered together with its k matched entrapments a potential original false discovery is equally likely to occupy each of the $k + 1$ possible ranks independently of how many of these $k + 1$ are reported discoveries and of the total number of discoveries K_{α} (Assumption 3). See also Assumption 3a and the discussion following it for common use cases. For $k = 1$ this is the paired estimation, Equation (4).

Table 1: **Summary of entrapment estimation methods discussed in this section.** Note that the required assumptions for the upper bound methods are variants of the equal chance assumption that TDC relies on: an incorrect discovery is equally likely to be a decoy or a target. Notably, the entrapment estimation does not require independence between the false discoveries, which TDC requires and which makes PSM-level FDR control particularly challenging.

2- Following the same logic, the authors refer a couple of time to the entrapment query protocol which Madej and Lam warned against. I understand their general setting is not incompatible with entrapment query, but the way it is phrased looks like the authors would like to re-open the door Madej and Lam tentatively closed. In my opinion, both views are however not incompatible: Madej and Lam warned about issues with the previous implementations, while the authors explains it could be implemented correctly in the future. A

nice way to reconcile this would be to underline that although theoretically valid, implementing entrapment query protocols that provably fulfill the assumptions summarized in the table resulting from my previous comment is challenging and remains an unaddressed question.

We agree with the reviewer, and we have already tried to address this point as part of our Discussion analysis of Madej and Lam's claim: "Second, if we believe for example that Assumption 2a holds, then the combined estimation method is valid regardless of which type of decoys the analysis tool uses to control the FDR." However, to further clarify the point we modified the sentence three paragraphs after the latter paragraph to read

Regardless, it is clear that the topic of entrapment expansion and its impact on the validity of our entrapment assumptions such as Assumption 1 merits further research.

3- In the previous round, I was partly unsatisfied with my lack of advice about a better display of the practical consequences of Table 1 results (essentially, that DIA is considered more sensitive than DDA, especially in SC but also in bulk analyses, partly because the FDR is not properly control; a point also raised by another reviewer) but in between, I got an idea: So far, the authors focused on comparing the claimed FDR control and the one obtained with FDRBench (First paragraph of p.10 and Supp mat S5). In themselves, the figures are striking, at least for a biostatistician, but my general feeling is that many wet-lab researcher would say "ok but shifting an FDR threshold from 1% to 2% is not a big deal, we essentially need a community standart, I am not sure it will change my understanding of the sample". Conversely, if the authors pinpoint that those FDR shift leads to an artificial increment of X% of identified peptides or proteins, then things become concrete as any researcher comparing DDA and DIA will be able to evaluate which part of the DIA improved sensitivity is real, and which part was wrongly claimed as a result of an erroneous FDR control procedure. I find it would give ground to the assertions that the lack of statistical robustness of DIA tools have consequences as the authors claim in the introduction.

As suggested by the reviewer we added a new plot (Figure 1b here, 4b in the paper), accompanied by the following paragraph dedicated to showing the estimated inflation rate due to the failure to control the FDR.

To investigate the practical consequences of erroneous FDR control, we compared the number of discoveries reported at the 1% FDR threshold by DIA-NN to the number of discoveries we get if we use the paired method to guide our cutoff. Specifically, for each of the ten datasets analyzed in Table 2 we asked how many more discoveries DIA-NN reports at the 1% FDR threshold relative to how many discoveries we get when the (paired) entrapment estimated FDP is at 1%. As shown in Figure 1b, when the estimated FDP is in the 1%–2% range, we see an estimated inflation of up to 6.7% in the number of discoveries at the precursor level and up to 4.7% at the protein level. In the case of the single cell proteomics dataset (1cell-eclipse), the estimated inflation rate is up to 48.3% at the precursor level and 44.2% at the protein level.

Minor:

- "Applying our entrapment procedures to DIA analysis tools we demonstrate that their FDR control at the peptide or precursor level can be questionable, and it is often questionable at the protein level." The result summary could be more nuanced: some FDR control are questionable, others are clearly invalid.

We modified the relevant sentence to

Applying our entrapment procedures to DIA analysis tools, we demonstrate that their FDR control at the peptide or precursor level occasionally seems to fail and typically fails at the protein level.

- when conditional probabilities involves cardinality, the equations tend to be less easy to read. Would the authors mind keeping "|" for conditioning and use an alternative notation (like "Card()") or "#[]" or whatever) for cardinality?

Done.

Figure 1: **Entrapment evaluation of the FDR control of DIA analysis tools.** (a) FDR control evaluation on the human-lumos DIA dataset. DIA-NN, EncyclopeDIA and Spectronaut were applied to the human-lumos DIA dataset using shuffled entrapment with $r = 1$. The top row shows the precursor or peptide-level estimated FDPs, and the bottom row shows the corresponding protein-level analysis. In the Spectronaut precursor-level plot, the x axis was set to show the maximum FDR threshold reported by the tool, which was less than the 1% threshold set in the analysis. The dashed vertical lines are at the 1% FDR threshold, as are the numbers reported in text in the panels. (b) Comparing the number of precursor and protein discoveries reported at the 1% FDR threshold by DIA-NN to the inferred number corresponding to the 1% entrapment-estimated FDP (paired method). The numbers at the top are the entrapment-estimated FDPs using the paired method at 1% FDR threshold. The three numbers on each bar are the estimated inflation rate, the reported number of discoveries (n_1) at 1% FDR threshold, and the entrapment inferred number of discoveries (n_2). The estimated inflation rate (Y axis) is calculated as $100\% \cdot (n_1 - n_2) / n_2$.

Reviewer 2

The revised manuscript has been much improved in terms of clarity and rigor. I would say that it is now up to the high standard of Nature Methods in those aspects, and my only reservation now is whether the focus and the scope is suitable for this journal.

We thank the reviewer for their positive assessment of our manuscript. We address their reservations below.

I would not object to this paper being published in its current form. However, I still believe that a heavier emphasis on DIA will make the paper more impactful, as the unresolved problem of error control in DIA is perhaps the biggest issue in our field right now. It would have been nice to dig a bit deeper and drive the point home, and organize the paper's overall message around its most important finding. Challenging the accuracy of FDR estimates of those widely used DIA tools is the big deal here. But where is the word "DIA" in the paper title?

As it is now, the readers of this paper would spend most of the time reading about the various variants of entrapment-based FDR estimation/validation methods, which, let's be honest, is just too technical for the broader audience of proteomics practitioners. I am afraid that the extremely long setting of the stage will turn off most readers before it gets to the key message of the paper. If the paper were all about comparing different ways of calculating FDP from entrapment searches, and concluding that they mostly work for DDA, it should not be published in Nature Methods. (I imagine that even the authors would agree with me on that.) In other words, the DIA part is the main reason why we are all excited about the work and feel that it deserves a bigger platform.

We agree that the problematic FDR control in the DIA setting is a major finding, and we believe that the new Figure 4b (1b here) helps to further drive home this message. However, the reason we were able to make this discovery was our realization that the field is fundamentally confused about how entrapment should be done. Our view is that, sooner or later, the problems with DIA FDR control will be fixed in the relevant tools. From that point onward, the lasting message of this paper will be the laying of a theoretical foundation for entrapment and advising the field how entrapment should and should not be done. Thus, we are inclined not to refocus more around the DIA results, especially given that we are advertising them quite clearly in the abstract, introduction and discussion.